# Attention modulates human visual responses to objects by tuning sharpening

**Narges Doostani[1,2]\*, Gholam-Ali Hossein-Zadeh[1,3], Radoslaw M Cichy[2], Maryam Vaziri-Pashkam[4,5]\***

[1]School of Cognitive Sciences, Institute for Research in Fundamental Sciences (IPM), Tehran, Iran; [2]Department of Education and Psychology, Freie Universität Berlin, Berlin, Germany; [3]School of Electrical and Computer Engineering, College of Engineering, University of Tehran, Tehran, Iran; [4]Department of Psychological and Brain Sciences, University of Delaware, Newark, United States; [5]Laboratory of Brain and Cognition, National Institute of Mental Health, Bethesda, United States

## eLife Assessment

This **valuable** study has the potential to shed mechanistic light on how attention mechanisms that influence competition between multiple visual stimuli are modulated by the relative neural similarity of these stimuli. The study provides **convincing** data that will also be used for future modeling efforts. The study will be of interest to researchers working on the neural basis of visual attention.

**\*For correspondence:**
narges.doostani.d@gmail.com
(ND);
mvaziri@udel.edu (MV-P)

**Competing interest:** The authors declare that no competing interests exist.

**Abstract** Visual stimuli compete with each other for cortical processing and attention biases this competition in favor of the attended stimulus. How does the relationship between the stimuli affect the strength of this attentional bias? Here, we used functional MRI to explore the effect of target-distractor similarity in neural representation on attentional modulation in the human visual cortex using univariate and multivariate pattern analyses. Using stimuli from four object categories (human bodies, cats, cars, and houses), we investigated attentional effects in the primary visual area V1, the object-selective regions LO and pFs, the body-selective region EBA, and the scene-selective region PPA. We demonstrated that the strength of the attentional bias toward the target is not fixed but decreases with increasing target-distractor similarity. Simulations provided evidence that this result pattern is explained by tuning sharpening rather than an increase in gain. Our findings provide a mechanistic explanation for the behavioral effects of target-distractor similarity on attentional biases and suggest tuning sharpening as the underlying mechanism in object-based attention.

## Introduction

Everyday visual scenes typically contain a large number of stimuli. Since processing all the incoming information is impossible due to the brain's limited neural resources, different stimuli compete for cortical representation and processing (*Desimone and Duncan, 1995*; *Kastner et al., 1998*; *Reynolds et al., 1999*; *Beck and Kastner, 2005*; *Reddy et al., 2009*; *McMains and Kastner, 2011*). This competition can be biased by the top-down signal of attention to enhance the parts of the input that are most relevant to the task at hand (*Moran and Desimone, 1985*; *Desimone and Duncan, 1995*; *Reynolds et al., 1999*). Evidence from electrophysiology and fMRI studies have demonstrated the role of attention in biasing the competition by enhancing the response related to the attended stimulus by approximately 30% compared to its response when unattended, in both electrophysiology

studies of the monkey brain (*Treue and Maunsell, 1996*; *Reynolds et al., 1999*; *Treue and Maunsell, 1999*; *Reynolds and Desimone, 2003*; *Fallah et al., 2007*) and fMRI studies of the human brain (*Reddy et al., 2009*).

Competition and attentional bias likely depend on the nature of the visual scenes rather than being universally uniform. Behavioral studies indicate that the competition between stimuli is content-dependent (*Cohen et al., 2014*), with higher competition between stimuli that are located closer to each other (*Franconeri et al., 2013*), or between stimuli with more similar cortical representation patterns (*Cohen et al., 2014*; *Cohen et al., 2017*). This suggests that the attentional bias might also be affected by the relationship between the competing stimuli, such as the similarity of their cortical representation. Furthermore, behavioral studies on the effect of target-distractor similarity on performance have proposed that lower performance for more similar target-distractor pairs is due to the fact that the neural resources needed for detailed processing are shared to a greater extent (*Cohen et al., 2014*). However, a direct neuroscientific investigation of how target-distractor similarity affects visual representations, and a mechanistic explanation of how shared resources affect attentional biases are missing.

Here, we investigated the impact of similarity in cortical representation on attentional bias and the underlying mechanism with empirical and theoretical tools. First, using functional MRI and uni- as well as multivariate analysis, we investigated how the top-down effect of attention varies as target-distractor similarity changes for multiple presented objects. Specifically, we found that the strength of the attentional bias towards the target decreases with increasing target-distractor similarity in cortical representation.

Second, using simulations of neuronal populations we determined how this effect arises from attentional enhancement of neural responses. We considered two known mechanisms through which attention affects neural firing rate: response gain and tuning sharpening. The response gain model predicts a multiplicative scaling of responses through which neural responses are increased by a gain factor (*McAdams and Maunsell, 1999*; *Reynolds and Chelazzi, 2004*). The tuning sharpening model, instead, proposes that attentional enhancement depends on the neuronal tuning for the attended stimulus, leading to an increase in response for optimal stimuli, and little increase in response or even response suppression for non-optimal stimuli (*Martinez-Trujillo and Treue, 2004*). We find that the empirically-observed relationship between attentional enhancement and target-distractor similarity are predicted by the tuning sharpening model, but not the response gain model.

Together, our results show that attentional enhancement is dependent on the similarity between the target and the distractor in neural representation, and a more similar distractor causes the target to receive less attentional bias in the competition. Moreover, these results suggest tuning sharpening as the underlying mechanism of attentional enhancement during object-based attention.

## Materials and methods
### Main experiment
### Participants
17 healthy human participants (nine females, age: mean ± s.d.=29.29 ± 4.5 y) with normal or corrected-to-normal vision took part in the study. We estimated the number of participants conservatively based on the smallest amount of attentional modulation observed in our previous study (*Doostani et al., 2023*). For a medium effect size of 0.3 and a power of 0.8, we needed a minimum number of 16 participants. Participants gave written consent and received payment for their participation in the experiment. Data collection was approved by the Ethics Committee of the Institute for Research in Fundamental Sciences, Tehran.

The behavioral data for two participants was not correctly saved during the scanning due to technical problems. While we used the fMRI data of these two participants, all behavioral reports include the performance of the 15 participants for whom the behavioral data was properly saved.

### Stimulus set and experimental design
To determine the effect of target-distractor similarity on attentional modulation, we used object stimuli from four categories (human bodies, cars, houses, and cats). We included body and house categories because there are regions in the brain that are highly responsive and unresponsive to each of these

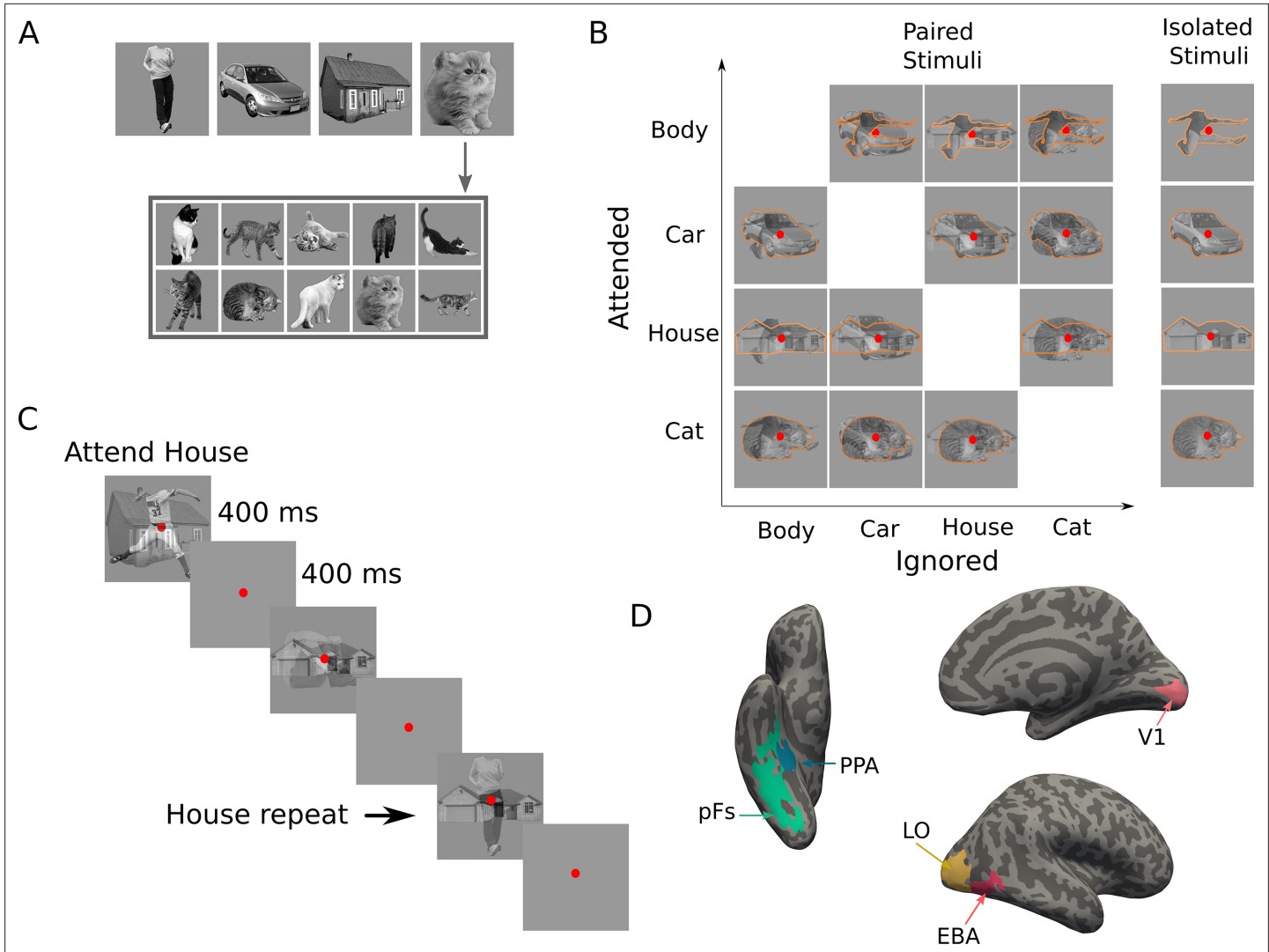

**Figure 1.** Stimuli, paradigm, and regions of interest. (**A**) Top images represent the four categories used in the main experiment: body, car, house, and cat. The stimulus set consisted of 10 exemplars from each category (here: cats), with exemplars differing in pose and 3D-orientation. (**B**) The experimental design comprised 16 task conditions (12 paired, four isolated). The 4×4 matrix on the left illustrates the 12 paired conditions, with the to-be-attended category (outlined in orange for illustration purposes, not present in the experiment) on the y-axis and the to-be-ignored category on the x-axis. The right column illustrates the four isolated conditions. (**C**) Experimental paradigm. A paired block is depicted with superimposed body and house stimuli. In this example block, house stimuli were cued as targets, and the participant responded on the repetition of the exact same house in two consecutive trials, as marked here by the arrow. (**D**) Regions of interest for an example participant; the primary visual cortex (V1), the object-selective regions lateral occipital cortex (LO) and posterior fusiform gyrus (pFs), the body-selective region extrastriate body area (EBA), and the scene-selective region parahippocampal place area (PPA).

The online version of this article includes the following figure supplement(s) for figure 1:

**Figure supplement 1.** Behavioral performance of the participants during the recording.

categories, which provided us with a range of responsiveness in the visual cortex. We chose the two remaining categories based on previous behavioral results to include categories that provided us with a range of similarities (*Xu and Vaziri-Pashkam, 2019*). Thus, for each category there was a range of responsiveness in the brain and a range of similarity with the other categories.

We presented stimuli from each category in semi-transparent form, either in isolation (isolated conditions), or paired with stimuli from another category (paired conditions). Thus, the experiment consisted of 16 conditions: 4 isolated conditions in which isolated stimuli from one of the four categories were presented, and 12 paired conditions (six category pairs × two attentional targets for each pair) in which a target stimulus from the cued category was superimposed with a distractor stimulus

from another category for all category combinations. *Figure 1B* depicts all stimulus conditions. We used isolated conditions to assess the similarity between different categories, and paired conditions to determine the effect of similarity in a category pair on attentional modulation.

The stimulus set consisted of gray-scaled images from the four object categories of human bodies, cats, cars, and houses, similar to stimuli used in previous studies (*Vaziri-Pashkam and Xu, 2017*; *Vaziri-Pashkam and Xu, 2019*; *Xu and Vaziri-Pashkam, 2019*). Each category consisted of 10 exemplars all varying in identity, 3D-orientation (for houses and cars), and pose (for bodies and cats, see *Figure 1A*).

Images were presented at the center of the screen on a gray background, subtending 10.2° of visual angle. A red fixation point subtending 0.45° of visual angle was presented at the center of the screen throughout the run (*Figure 1C*).

## Procedure

We used a blocked design for the main experiment. At the beginning of each block, participants were cued by a word to attend to either bodies, cars, houses, or cats. During the block, participants maintained attention on the images from the cued category, and performed a one-back repetition detection task on them by pressing the response button when the same stimulus from the attended category appeared in two consecutive trials. Repetition occurred two to three times at random times in each block. The experiment consisted of 16 block types, corresponding to the 16 task conditions (*Figure 1B*).

Each block lasted for 10 s, starting with the cue word presented for 1 s, followed by 1 s of fixation. Then, ten images from the cued category were presented in isolation or paired with ten images from another category. Each image was presented for 400 ms, followed by 400ms of fixation (*Figure 1C*). There were 8 s of fixation in between the blocks, and a final 8 s fixation after the last block.

We organized blocks in runs, each lasting 4 min 56 s. Each run started with 8 s of fixation followed by block presentations. The presentation order of the 16 task conditions was counterbalanced across each experimental run. 10 participants completed 16 runs and 7 participants completed 12 runs of the main experiment.

## Localizer experiments

Considering that we used object categories, we investigated five different regions of interest (ROIs): the object-selective areas lateral occipital cortex (LO) and posterior fusiform gyrus (pFs) as general object-selective regions, the body-selective extrastriate body area (EBA) and the scene-selective parahippocampal place area (PPA) as regions that are highly selective for specific categories, and the primary visual cortex (V1) as a control region. We chose these regions because they could all be consistently defined in both hemispheres of all participants and included a large number of voxels. To define these ROIs, each participant completed four localizer runs described in detail below.

## Early visual area localizer

We used meridian mapping to localize the primary visual cortex V1. Participants viewed a black-and-white checkerboard pattern through a 60-degree polar angle wedge aperture. The wedge was presented either horizontally or vertically. Participants were asked to detect luminance changes in the wedge in a blocked-design paradigm. Each run consisted of four horizontal and four vertical blocks, each lasting 16 s, with 16 s of fixation in between. A final 16 s fixation followed the last block. Each run lasted 272 s. The order of the blocks was counterbalanced within each run. Participants completed two runs of this localizer.

## Category localizer

We used a category localizer to localize the cortical regions selective to scenes (PPA), bodies (EBA), and objects (LO, pFs). In a blocked-design paradigm, participants viewed stimuli from the five categories of faces, scenes, objects, bodies, and scrambled images. The stimuli differed from those used in the main experiment. Each localizer run contained two 16 s blocks of each category, with the presentation order counterbalanced within each run. An 8 s fixation period was presented at the beginning, in the middle, and at the end of the run. In each block, 20 stimuli from the same category were presented. Stimuli were presented for 750 ms followed by 50 ms of fixation on a gray background

screen. Participants were asked to maintain their fixation on a red circle at the center of the screen throughout and press a key when they detected a slight jitter in the stimuli that happened two to three times per block. Each run lasted 344 s. Participants completed two runs of this localizer.

## Stimulus presentation inside the scanner

We back-projected the stimuli onto a screen positioned at the rear of the magnet using an LCD projector with a refresh rate of 60 Hz and a spatial resolution of 768 × 1024. Participants observed the screen through a mirror attached to the head coil.

## MRI data acquisition

We recorded the data of 10 participants using the Siemens 3T Tim Trio MRI system with a 32-channel head coil at the Institute for Research in Fundamental Sciences (IPM). We collected the data of seven additional participants on a Siemens Prisma MRI system using a 64-channel head coil at the National Brain-mapping Laboratory (NBML). For each participant, we performed a whole-brain anatomical scan using a T1-weighted MPRAGE sequence. For the functional scans, including the main experiment and the localizer experiments, we acquired 33 slices parallel to the AC-PC line using T2*-weighted gradient-echo echo-planar imaging (EPI) sequences covering the whole brain (TR=2 s, TE=30 ms, flip angle=90°, voxel size=3 × 3 × 3 mm$^3$, matrix size = 64 × 64).

## fMRI data preprocessing

We performed fMRI data analysis using FreeSurfer (https://surfer.nmr.mgh.harvard.edu), Freesurfer Functional Analysis Stream (*Dale et al., 1999*) and in-house MATLAB codes. fMRI data preprocessing steps included 3D motion correction, slice timing correction, and linear and quadratic trend removal. We performed no spatial smoothing on the data. We used a double gamma function to model the hemodynamic response function. We eliminated the first four volumes (8 s) of each run to avoid the initial magnetization transient. We compared the SNR values of the two groups of participants and observed no significant difference between these values ($ps > 0.34$, $ts < 0.97$).

## fMRI data analysis

For the main experiment, we performed a general linear model (GLM) analysis for each participant to estimate voxel-wise regression coefficients in each of the 16 task conditions. The onset and duration of each block were convolved with a hemodynamic response function and were then entered into the GLM as regressors. We also included movement parameters and linear and quadratic nuisance regressors in the GLM. We did not enter the cue to the GLM as a predictor. The obtained voxel-wise coefficients for each condition are thus related to the cue and the stimuli presented in that condition. We used these voxel-wise coefficients from the five ROIs as the basis for all further analyses.

For the early visual area localizer experiment, we estimated voxel regression coefficients in each of the two conditions (i.e. vertical and horizontal wedge) using a separate GLM. After convolving with a hemodynamic response function, the onset and duration of each block were entered to the GLM as regressors of interest. We also included movement parameters and linear and quadratic nuisance regressors in the GLM. We used the obtained coefficients to define the V1 ROI.

For the category localizer, we used another GLM to estimate voxel-wise regression coefficients in the five task conditions (i.e. faces, scenes, objects, bodies, and scrambled images). The GLM procedure was similar to the other two experiments. We then used these estimated coefficients to define the LO, pFs, EBA, and PPA ROIs.

### Definition of ROIs

We determined the V1 ROI using a contrast of horizontal versus vertical polar angle wedges that reveals the topographic maps in the occipital cortex (*Sereno et al., 1995*; *Tootell et al., 1998*). To define the object-selective areas LO in the lateral occipital cortex and pFs in the posterior fusiform gyrus (*Malach et al., 1995*; *Grill-Spector et al., 1998*), we used a contrast of objects versus scrambled images. We selected the active voxels in the lateral occipital and ventral occipitotemporal cortex as LO and pFS, respectively, following the procedure described by *Kourtzi and Kanwisher, 2000*. We used a contrast of scenes versus objects for defining the scene-selective area PPA in the parahippocampal gyrus (*Epstein et al., 1999*), and a contrast of bodies versus objects for defining the

body-selective area EBA in the lateral occipitotemporal cortex (*Downing et al., 2001*). We thresholded the activation maps for both the early visual localizer and the category localizer at $p < 0.001$, uncorrected. We selected this threshold to allow for the selection of a reasonable number of voxels in each hemisphere across all participants.

## Univariate fMRI analysis

We first used a univariate analysis to determine the effect of attention for different category pairs. Using the voxel-wise coefficients of the isolated conditions associated with each category, we examined the relative response of each voxel to the two categories for each category pair. This relative response determined which of the two categories was more preferred by the voxel. Therefore, for each category pair and each voxel, the category that elicited a higher response in the isolated condition was assigned the relatively more preferred category ($M$) label and the other the relatively less preferred category ($L$) label.

### Univariate distance based on the isolated conditions

We had six pairs of categories: Body-Car, Body-House, Body-Cat, Car-House, Car-Cat, and House-Cat. As a measure of the difference between the response evoked by each of the two categories in a pair, we defined a univariate distance. We calculated the univariate distance for each pair of categories simply as the difference in voxel responses of the two isolated conditions (*Equation 1*):

$$Univariate\ distance = R_{M^{at}} - R_{L^{at}} \qquad (1)$$

Here, $R$ denotes the average voxel response across runs, and the subscripts $M$ and $L$ denote the presence of the more preferred and the less preferred stimuli, respectively. The superscript $at$ denotes the attended stimulus. Note that in the isolated conditions, the presented stimulus was always attended. Thus, $R_{M^{at}}$ is the average response related to the isolated preferred stimulus, and $R_{L^{at}}$ is the average response to the isolated less preferred stimulus. For example, the Body-Car univariate distance was assessed by $R_{Body^{at}} - R_{Car^{at}}$ for voxels more responsive to bodies than cars, and by $R_{Car^{at}} - R_{Body^{at}}$ for voxels more responsive to cars than bodies. Thus, according to this measure, two categories that elicited closer responses had less univariate distance, indicating more similarity in univariate response between the two categories.

### Univariate effect of attention based on the paired conditions (Univariate shift)

For each of the six category pairs, we had two paired conditions, in which stimuli from both categories were presented, but with attention directed to either one or the other category (for example, $Body^{at}Car$ and $BodyCar^{at}$ conditions for the Body-Car pair, with the superscript $at$ denoting the to-be-attended stimulus). Since these paired conditions differed only in the attentional target and not in the stimuli, any difference observed in cortical response can be ascribed to the shift in attention (*O'Craven et al., 1999*; *Ni et al., 2012*; *Vaziri-Pashkam and Xu, 2017*; *Doostani et al., 2023*). We thus defined the univariate shift for each pair of categories as the change in response when attention shifted from the more preferred stimulus to the other:

$$Univariate\ shift = R_{M^{at}L} - R_{ML^{at}} \qquad (2)$$

Here, $R_{M^{at}L}$ denotes the response related to the paired condition with attention directed to the more preferred category, while $R_{ML^{at}}$ is the elicited response when attending to the less preferred category in the pair. For example, considering the Body-Car pair, we assessed the univariate shift by $R_{Body^{at}Car} - R_{BodyCar^{at}}$ for a voxel preferring bodies to cars, and by $R_{BodyCar^{at}} - R_{Body^{at}Car}$ for a voxel preferring cars to bodies.

## Multivariate pattern analysis

To determine the effect of attention at the multivariate level, and to examine the attentional bias that the representation of the stimulus in each category pair receives, we used a multivariate pattern analysis. Here, rather than comparing the mean values of voxel-wise coefficients in each ROI, we instead considered the ROI response pattern in each condition as a response vector, with the voxel-wise coefficients as its elements. Therefore, we had 16 response vectors, one for each task condition, in

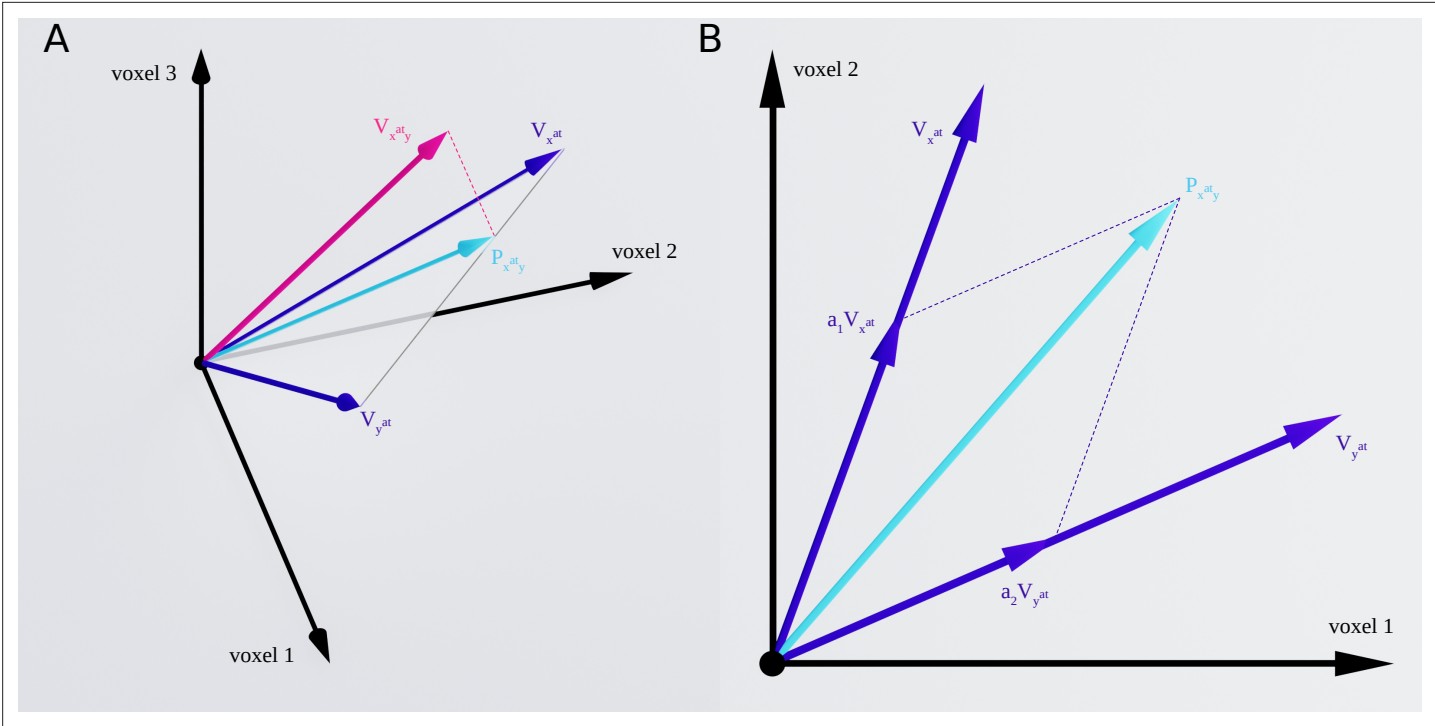

**Figure 2.** Response vectors related to $x$ and $y$ stimuli in isolated and paired conditions. (**A**) Example illustration of the isolated and paired responses in three-dimensional space. $V_{x^{at}}$ and $V_{y^{at}}$ denote the response vectors related to isolated $x$ and isolated $y$ conditions and $V_{x^{at}y}$ illustrates the response in the paired condition with attention directed to stimulus $x$. The paired response is projected on the plane defined by the two isolated responses $V_{x^{at}}$ and $V_{y^{at}}$. This projection is illustrated by $P_{x^{at}y}$. (**B**) Two-dimensional illustration of the plane defined by the two isolated response vectors $V_{x^{at}}$ and $V_{y^{at}}$, along with the paired response vectors $V_{x^{at}}$ and $V_{y^{at}}$ and $P_{x^{at}y}$ as the projection of the paired response, $V_{x^{at}y}$, on the plane. We calculated the weight of the two isolated responses in the paired response using multiple regression, with the weights of $V_{x^{at}}$ and $V_{y^{at}}$ shown as $a_1$ and $a_2$, respectively.

each ROI. Similar to the univariate analysis, we used the responses in the isolated conditions to assess category distance, and the responses in the paired conditions to evaluate the effect of attention.

*Figure 2A* illustrates the response vectors for two stimulus categories (here termed $x$ and $y$) in the isolated conditions, as well as the response vector to the paired condition with attention directed to $x$. The three vectors $V_{x^{at}}$, $V_{y^{at}}$, and $V_{x^{at}y}$ illustrate the response patterns of these three conditions, with $V$ representing the response vector in an ROI, subscripts $x$ and $y$ denoting the presence of the $x$ and $y$ stimuli, respectively, and the superscript $at$ denoting the attended stimulus. Therefore, $V_{x^{at}}$ represents the response vector related to the isolated $x$ condition (in which $x$ was automatically attended), and $V_{x^{at}y}$ represents the response vector related to the paired $xy$ condition with attention directed to the $x$ stimulus. The projection of the paired-condition vector $V_{x^{at}y}$ onto the plane defined by the two isolated responses $V_{x^{at}}$ and $V_{y^{at}}$ is illustrated as $P_{x^{at}y}$. Using this projection vector, we calculate the weight of $V_{x^{at}}$ and $V_{y^{at}}$ in the paired response.

## Multivariate distance based on the isolated conditions

As illustrated in *Figure 2B*, the two isolated response vectors $V_{x^{at}}$ and $V_{y^{at}}$ have a certain distance because the response across the voxels varies for the two stimuli. For two stimuli that elicit more similar response patterns in an ROI, the isolated response vectors are closer to each other. Thus, we defined the multivariate distance between the two isolated response vectors $V_{x^{at}}$ and $V_{y^{at}}$ in each ROI using Pearson's correlation, as shown in *Equation 3*:

$$Multivariate\ distance = 1 - \rho(V_{x^{at}}, V_{y^{at}}) \tag{3}$$

where $V_{x^{at}}$ and $V_{y^{at}}$ represent the response vectors related to the isolated $x$ and $y$ conditions and $\rho$ denotes Pearson's correlation coefficient between the two response vectors. For stimuli with more similar response patterns, the correlation between their response vectors will be higher, leading to lower multivariate distance.

## Multivariate effect of attention based on the paired conditions (Attentional weight shift)

Similar to the isolated conditions, we considered the response pattern in the paired conditions as vectors, $V_{x^{at}y}$ and $V_{xy^{at}}$. We first projected the paired vectors on the plane defined by the isolated vectors (*Figure 2A*) and then determined the weight of each isolated vector in the projected vector (*Figure 2B*). Thus, the response vectors in the paired conditions can be written as the linear combination of the response vectors in the isolated conditions, with an error term denoting the deviation of the paired-condition responses from the plane defined by the isolated-condition responses (*Reddy et al., 2009*), as shown in *Equation 4a*:

$$V_{x^{at}y} = a_1 \cdot V_{x^{at}} + a_2 \cdot V_{y^{at}} + \epsilon_1 \tag{4a}$$

$$V_{xy^{at}} = b_1 \cdot V_{x^{at}} + b_2 \cdot V_{y^{at}} + \epsilon_2 \tag{4b}$$

Here, parameters $a_1$ and $a_2$ are the weights of the isolated $x$ and $y$ responses, respectively, when $x$ is attended, and parameters $b_1$ and $b_2$ are the respective weights of isolated $x$ and $y$ responses when $y$ is attended. $\epsilon_1$ and $\epsilon_2$ denote the error terms related to the deviation of the $V_{x^{at}y}$ and $V_{xy^{at}}$ from the $V_{x^{at}} - V_{y^{at}}$ plane, respectively. While this model has been previously called *weighted average* (*Reddy et al., 2009*), we chose the more general term *linear combination* because we did not impose any limits on the estimated weights of the two isolated responses in the paired response.

A higher $a_1$ compared to $a_2$ indicates that the paired response pattern is more similar to $V_{x^{at}}$ than to $V_{y^{at}}$, and vice versa. For instance, after calculating the weights of the Body and Car stimuli in the paired response related to the simultaneous presentation of both stimuli in the LO ROI, we obtain: $V_{Body^{at}Car} = 0.79V_{Body} + 0.31V_{Car}$, $V_{BodyCar^{at}} = 0.43V_{Body} + 0.68V_{Car}$ (See *Appendix 1—table 4* for the average weights of the two stimuli for all pairs in all ROIs). Note that these weights are averaged across participants. As can be observed, in the presence of both body and car stimuli, the weight of each stimulus is higher when attended compared to the case when it is unattended. In other words, when attention shifts from body to car stimuli, the weight of the isolated body response ($V_{Body^{at}}$) decreases in the paired response. We can, therefore, observe in this instance that the response in the paired condition is more similar to the isolated body response pattern when body stimuli are attended and more similar to the isolated car response pattern when car stimuli are attended.

In the presence of two stimuli, if attention could completely remove the effect of the unattended stimulus, the paired response would be the same as the response to the isolated attended stimulus. However, the information related to the unattended stimulus is not fully removed and attention has been shown to increase the weight of the response related to the attended stimulus in the paired response without completely removing the effect of the unattended stimulus (*Reddy et al., 2009*), as observed in the above example of the Body-Car pair. As shown here, even when body stimuli were attended, the effect of the unattended car stimuli was still present in the response, shown in the weight of the isolated car response (0.31). This weight increased when attention shifted towards car stimuli to a value of less than 1 (0.68 in the attended case), showing the effect of attention on the response. To examine whether this increase in the weight of the attended stimulus is constant or if it depends on the similarity of the two stimuli in cortical representation, we defined the weight shift as the multivariate effect of attention:

$$\Delta w = \frac{a_1}{a_1 + a_2} - \frac{b_1}{b_1 + b_2} \tag{5}$$

Here, $a_1$, $a_2$, $b_1$, and $b_2$ are the weights of the isolated responses, estimated using *Equation 4a*. We calculate the weight of the isolated $x$ response once when attention is directed towards $x$ ($a_1$), and a second time when attention is directed towards $y$ ($b_1$). In each case, we calculate the relative weight of the isolated $x$ in the paired response by dividing the weight of the isolated $x$ by the sum of weights of $x$ and $y$ ($a_1 + a_2$ when attention is directed towards $x$, and $b_1 + b_2$ when attention is directed towards $y$). We then define the weight shift, $\Delta w$, as the change in the relative weight of the isolated $x$ response in the paired response when attention shifts from $x$ to $y$. A higher $\Delta w$ for a category pair indicates that attention is more efficient in removing the effect of the unattended stimulus in the pair. We used relative weights as a normalized measure to compensate for the difference in the sum of weights for different category pairs. Thus, using the normalized measure, we calculated the share of each stimulus in the paired response. For instance, considering the Body-Car pair, the share of the body stimulus in

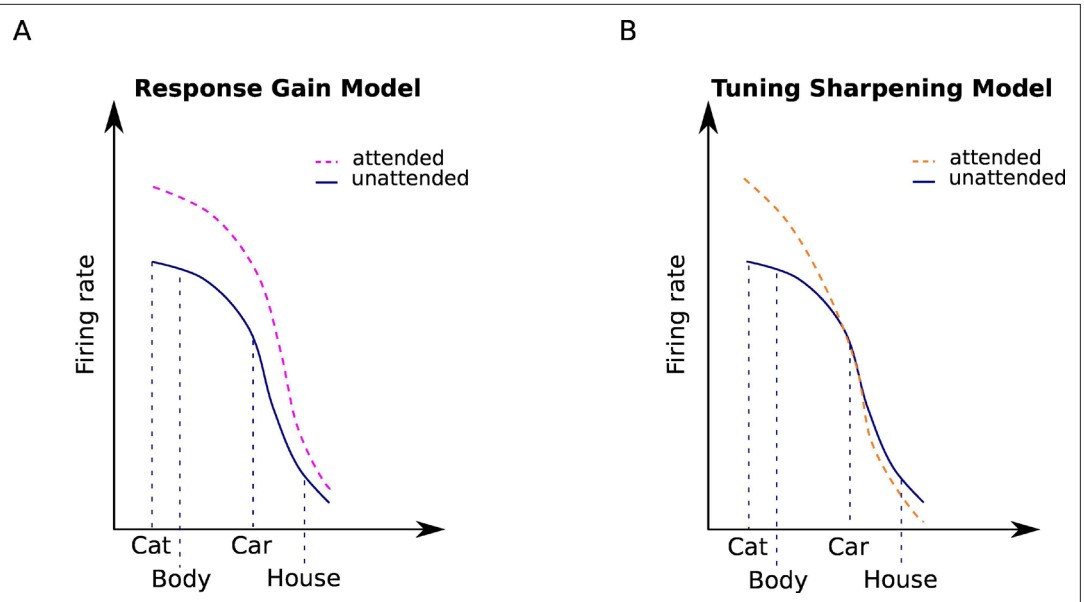

**Figure 3.** Attentional modulation by the response gain and tuning sharpening models. We illustrate the models here for the example of a neuron with high selectivity for cat stimuli. Solid curves denote the response to unattended stimuli and dashed curves denote the response to attended stimuli. (**A**) According to the response gain model, the response of the neuron to attended stimuli is scaled by a constant attention factor. Therefore, the response of the cat-selective neuron to an attended stimulus is enhanced to the same degree for all stimuli. (**B**) According to the tuning sharpening model, the response modulation by attention depends on the neuron's tuning for the attended stimulus. Therefore, for optimal and near-optimal stimuli such as cat and body stimuli the response is highly increased, while for non-optimal stimuli such as houses, the response is suppressed.

the paired response was equal to 0.72 and 0.38, when body stimuli were attended and unattended, respectively. We then calculated the change in the share of each stimulus caused by the shift in attention using a simple subtraction (*Equation 5*: $\Delta w = 0.34$ for the above example of the Body-Car pair in LO) and used this measure to compare between different pairs.

## Simulations

We investigated the mechanisms underlying the observed effect of stimulus similarity on attentional enhancement using simulations. To examine which attentional mechanism leads to the effects observed in the empirical data, we generated the neural response to unattended object stimuli as a baseline response in the absence of attention, using the data reported by neural studies of object recognition in the visual cortex (*Ni et al., 2012*; *Bao and Tsao, 2018*). Then, using an attention parameter for each neuron and different attentional mechanisms, we simulated the response of each neuron to the different task conditions in our experiment. Finally, we assessed the population response by averaging neural responses. We considered two models for attentional enhancement: a response gain model (*McAdams and Maunsell, 1999*; *Reynolds and Chelazzi, 2004*) and a tuning sharpening model (*Martinez-Trujillo and Treue, 2004*; *Ling et al., 2009*).

According to the response gain model, attention to an object multiplicatively increases neural responses to that object (*Figure 3A*). For instance, for a body-selective neuron, this mechanism can be implemented using *Equation 6a*:

$$R_{Body^{at}} = \beta \cdot R_{Body} \tag{6a}$$

$$R_{Car^{at}} = \beta \cdot R_{Car} \tag{6b}$$

Here, $R_{Body}$ is the neuron's response to an ignored body stimulus, and $R_{Body^{at}}$ is the response of the neuron to the attended body stimulus, which is enhanced by the attention factor, β and $R_{Car^{at}}$ in *Equation 6b* denote the response of the same body-selective neuron to an ignored and an attended car stimulus, respectively. The response gain model posits that attention to either stimulus enhances the

response of the neuron by the same attention factor. This multiplicative scaling preserves the shapes of the neurons' tuning curves (**Treue and Martínez Trujillo, 1999**; **McAdams and Maunsell, 1999**).

In contrast, according to the tuning sharpening model, attention to an object increases neural responses relative to their responsiveness to that object (**Figure 3C**). Therefore, while the response of a neuron is substantially enhanced when an optimal stimulus is attended, its response to an attended non-optimal stimulus is increased to a lesser degree or even decreased. The tuning sharpening model thus predicts a sharpening of the neurons' tuning curve with attention (**Ling et al., 2009**).

We implemented this mechanism using **Equation 7a**:

$$R_{Body^{at}} = \beta \cdot s_1 \cdot R_{Body} \tag{7a}$$

$$R_{Car^{at}} = \beta \cdot s_2 \cdot R_{Car} \tag{7b}$$

$$s_1 = \frac{R_{Body}}{R_{max}} \tag{7c}$$

$$s_2 = \frac{R_{Car}}{R_{max}} \tag{7d}$$

In the above equations, $R_{Body}$, $R_{Car}$, $R_{Body^{at}}$, and $R_{Car^{at}}$ denote the neuron's response to the unattended body, unattended car, attended body, and attended car stimuli, respectively. Parameters $s_1$ and $s_2$ denote the degree of the neuron's selectivity to body and car stimuli, respectively. Parameter β is the attention factor. $R_{max}$ is the response of the neuron to its optimal stimulus.

We simulated the action of the response gain model and the tuning sharpening model using numerical simulations. We composed a neural population of $4 \times 10^5$ neurons in equal proportions body-, car-, cat- or house-selective. Each neuron also responded to object categories other than its preferred category, but to a lesser degree and with variation. We chose neural responses to each stimulus from a normal distribution with the mean of 30 spikes/s and a standard deviation of 10 and each neuron was randomly assigned an attention factor in the range between 1 and 10 using a uniform distribution. These values are comparable with the values reported in neural studies of attention and object recognition in the ventral visual cortex (**Ni et al., 2012**; **Bao and Tsao, 2018**). We also added Poisson noise to the response of each neuron (**Britten et al., 1993**), assigned randomly for each condition of each neuron.

Attention was implemented according to the above equations. Using **Equations 6a and 7a**, we calculated the response of each neuron to the same 16 conditions as our main fMRI experiment. Then, we randomly chose 1000 neurons with similar selectivity from the population, and averaged their responses to make up a voxel.

We modeled two neural populations: a general object-selective population in which each voxel shows preference to a particular category and voxels with different preferences are mixed in with each other (similar to LO and pFS), and a category-selective population in which all voxels have a similar preference for a particular category (similar to EBA and PPA). Finally, we performed the same univariate and multivariate analyses as those used for the fMRI data to compare the predictions of each model with the observed data.

## Results
### Behavioral results

Participants performed a one-back repetition detection task to maintain attention toward the cued stimuli. Detection rate in each experimental run was checked during the scan to ensure that participants followed the instructions. Participants had an average detection rate of 90.49% across all runs, confirming effective attention towards the cued stimuli (**Figure 1—figure supplement 1**). As expected, the average detection rate in the isolated conditions (94.82% ± 0.046) was significantly higher than in the paired conditions (89% ± 0.07, with $t(14) = 7.2$ and $p < 0.0001$), since detecting a repetition in the superimposed case was more difficult.

## The effect of attention varies dependent on the target-distractor difference in response

We considered the effect of attention in five ROIs: the primary visual cortex V1, the object-selective regions LO and pFs, the body-selective region EBA, and the scene-selective region PPA. We obtained the voxel-wise responses through a GLM in those ROIs for all task conditions, consisting of four isolated conditions (blocks with isolated stimuli from one category) and 12 paired conditions (blocks with superimposed stimuli from two categories, see *Figure 1B*). There were six combinations of category pairs: Body-Car, Body-House, Body-Cat, Car-House, Car-Cat and House-Cat. For each voxel, we determined its relative preference for the two categories of each category pair, based on its response to the two categories in isolation. Thus, for each pair, one category was labeled as the more preferred category ($M$), and the other as the less preferred category ($L$). Considering the isolated and paired conditions related to each category pair, we hereafter refer to the conditions related to each category pair as $M^{at}$, $M^{at}L$, $ML^{at}$, and $L^{at}$, with $M$ and $L$ denoting the more preferred and the less preferred categories for each voxel, and the superscript $at$ denoting the attended stimulus.

For instance, for the Body-Car pair, for a voxel that showed a higher response to body stimuli than to car stimuli, the four associated conditions related to the pair were referred to as $M^{at}$ (attended body stimuli), $M^{at}L$ (attended body stimuli paired with ignored car stimuli), $ML^{at}$ (attended car stimuli paired with ignored body stimuli), and $L^{at}$ (attended car stimuli). If the same voxel was more responsive to

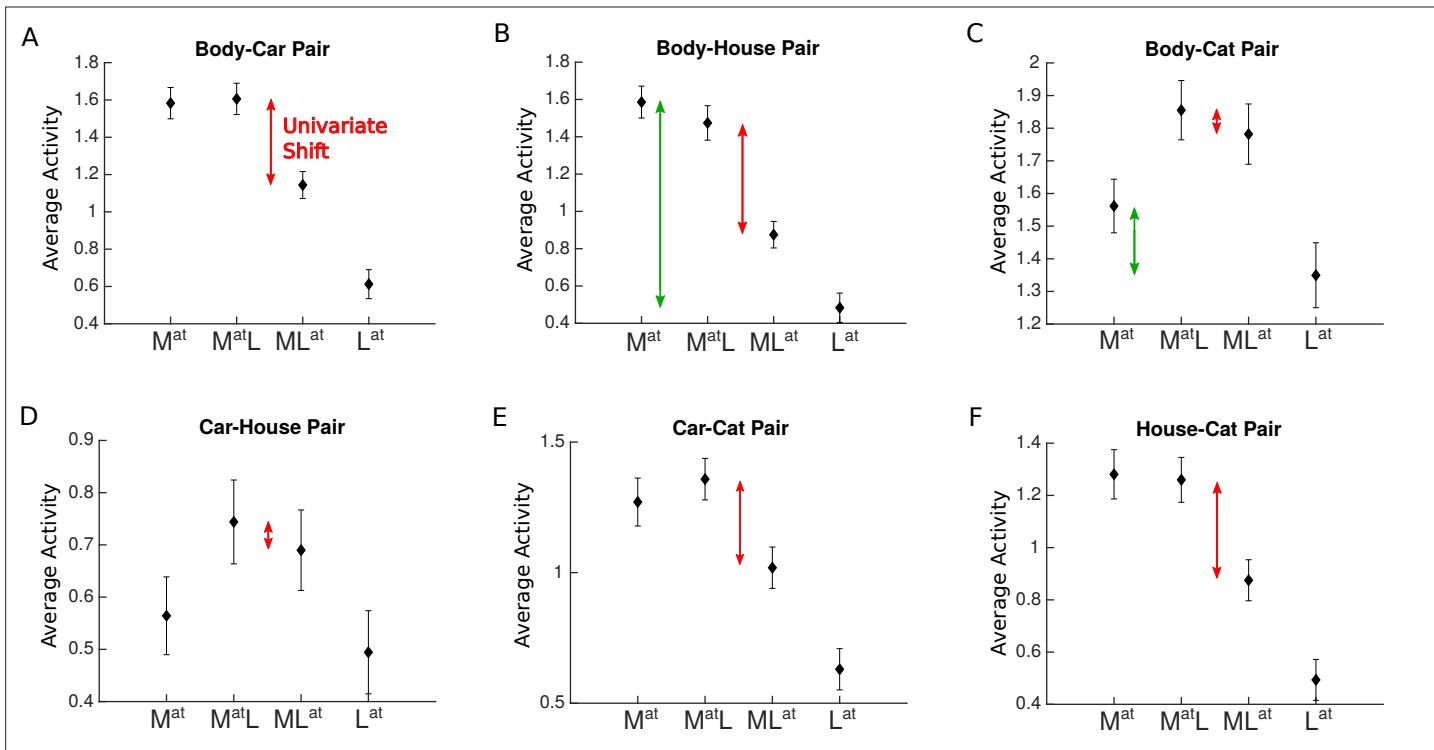

**Figure 4.** Average voxel response in the extrastriate body area (EBA) for each pair of stimulus categories. The x-axis labels represent the four conditions related to each category pair, $M^{at}$, $M^{at}L$, $ML^{at}$, $L^{at}$, with $M$ and $L$ denoting the presence of the more preferred and the less preferred category and the superscript $at$ denoting the attended category. For instance, $M^{at}$ refers to the condition in which the more preferred stimulus was presented in isolation (and automatically attended), and $ML^{at}$ refers to the paired condition in which the less preferred stimulus was attended to. Red arrows in each panel illustrate the observed change in response (univariate shift) caused by the shift of attention from the more preferred to the less preferred stimulus. Green arrows in panels B and C illustrate the difference in the response to isolated stimuli. Error bars represent standard errors of the mean. N=17 human participants.

The online version of this article includes the following figure supplement(s) for figure 4:

**Figure supplement 1.** Average voxel response in the primary visual cortex (V1) for each pair of stimulus categories.

**Figure supplement 2.** Average voxel response in the lateral occipital cortex (LO) for each pair of stimulus categories.

**Figure supplement 3.** Average voxel response in the posterior fusiform gyrus (pFs) for each pair of stimulus categories.

**Figure supplement 4.** Average voxel response in the parahippocampal place area (PPA) for each pair of stimulus categories.

cats than bodies, then the four conditions related to the Body-Cat pair would be referred to as: $M^{at}$ (attended cat stimuli), $M^{at}L$ (attended cat stimuli paired with ignored body stimuli), $ML^{at}$ (attended body stimuli paired with ignored cat stimuli), and $L^{at}$ (attended body stimuli).

Note that the response in paired conditions can be higher or lower than the response to the isolated more preferred stimulus (condition $M^{at}$), depending on the voxel response to the two presented stimuli (see *Figure 4C–D*), as previously reported (*Doostani et al., 2023*). This is consistent with previous studies reporting the response to multiple stimuli to be higher than the average, but lower than the sum of the response to isolated stimuli (*Reddy et al., 2009*).

We next determined the amount of univariate shift for each category pair using the voxel-wise coefficients related to the two paired conditions, $M^{at}L$ and $ML^{at}$. As illustrated in *Figure 4*, we defined univariate shift for each category pair as the reduction in response when attention shifted from the $M$ category to the $L$ category in the presence of both stimuli (*O'Craven et al., 1999*; *Ni et al., 2012*; *Vaziri-Pashkam and Xu, 2017*; *Doostani et al., 2023*).

We observed a significant univariate shift when attention shifted from the $M$ stimulus to the $L$ stimulus for all pairs in the higher-level ROIs ($ts > 3$, $ps < 0.04$, *corrected*) except for the Body-Car, Body-Cat, and Car-Cat pairs in PPA ($ts < 2$, $ps > 0.3$, *corrected*) and the Car-House pair in EBA ($t(16) = 1$, $p = 0.9$, *corrected*). In V1, we observed no significant univariate shift for any pairs ($ts < 2.5$, $ps > 0.1$, *corrected*) except for the Body-Car pair ($t(16) = 3.8$, $p < 0.01$, *corrected*). Thus, the observed effect was limited to higher-level visual areas. Since the presented stimuli were the same in both conditions, this effect is caused by the shift in attention. It is important to note that since the cue was not separately modeled in the GLM, the signals related to the cue and the stimuli were mixed. However, given that the cues were brief and presented in the form of words, they are unlikely to have an effect on the responses observed in the higher-level ROIs.

Closer comparison of the results suggests that for pairs with significant univariate shift, the shift is not uniform. Instead, it is greater for pairs in which the $M$ and $L$ stimuli elicited more different responses compared to pairs with $M$ and $L$ stimuli eliciting closer responses. For example, we observed a larger univariate shift for the Body-House pair (*Figure 4B*) compared to the Body-Cat pair (*Figure 4C*) in all ROIs ($ts > 4$, $ps < 0.001$, *Figure 4B–C*, compare the size of the red arrows) except for V1 ($t(16) = 0.65$, $p = 0.5$). Comparing the isolated responses for these two pairs, we observed that the difference between the response of the isolated Body and isolated House conditions was generally higher than the difference between the isolated Body and isolated Cat conditions in all ROIs ($ts > 4$, $ps < 0.001$, *Figure 4B–C*, compare the size of the green arrows).

To examine this relationship quantitatively for all category pairs, we used two approaches. First, in a univariate analysis using average voxel responses, we determined the relationship between the observed univariate shift and the difference in isolated responses. Next, in a multivariate pattern analysis, we considered the response patterns in each ROI and looked for the underlying basis of this effect of attention at the multivariate level. This analysis enabled us to determine whether the bias of attention on the representation of the attended stimulus differed for different category pairs.

## The univariate effect of attention decreases for target-distractor pairs that elicit closer responses

We first used a univariate analysis to determine the relationship between the univariate shift and category distance across pairings and in different ROIs. We split the fMRI data into two halves. Using the first half, we determined the voxel-wise $M$ and $L$ categories for each category pair. We then calculated the difference in the isolated response elicited by the two categories (univariate category distance) using the two isolated conditions $M^{at}$ and $L^{at}$ (*Equation 1*).

Then, using the second left-out part of the data, we assessed the univariate shift related to the pair as the amount of the reduction in response when attention shifted from the $M$ stimulus to the $L$ stimulus in the paired presentation of both stimuli (*Equation 2*). For instance, for the Body-Car pair and a voxel more responsive to bodies than cars, univariate category distance was calculated by $R_{Body^{at}} - R_{Car^{at}}$, and univariate shift was calculated by $R_{Body^{at}Car} - R_{BodyCar^{at}}$.

We observed a significantly positive correlation between univariate shift and category distance in all ROIs ($ts > 2.5$, $ps < 0.02$) except V1 ($t(16) = 0.56$, $p = 0.58$, see *Figure 5*). These results demonstrate that for stimuli that elicit more different responses, attention causes a greater response modulation, while the shift of attention between stimuli with more similar responses causes little response change.

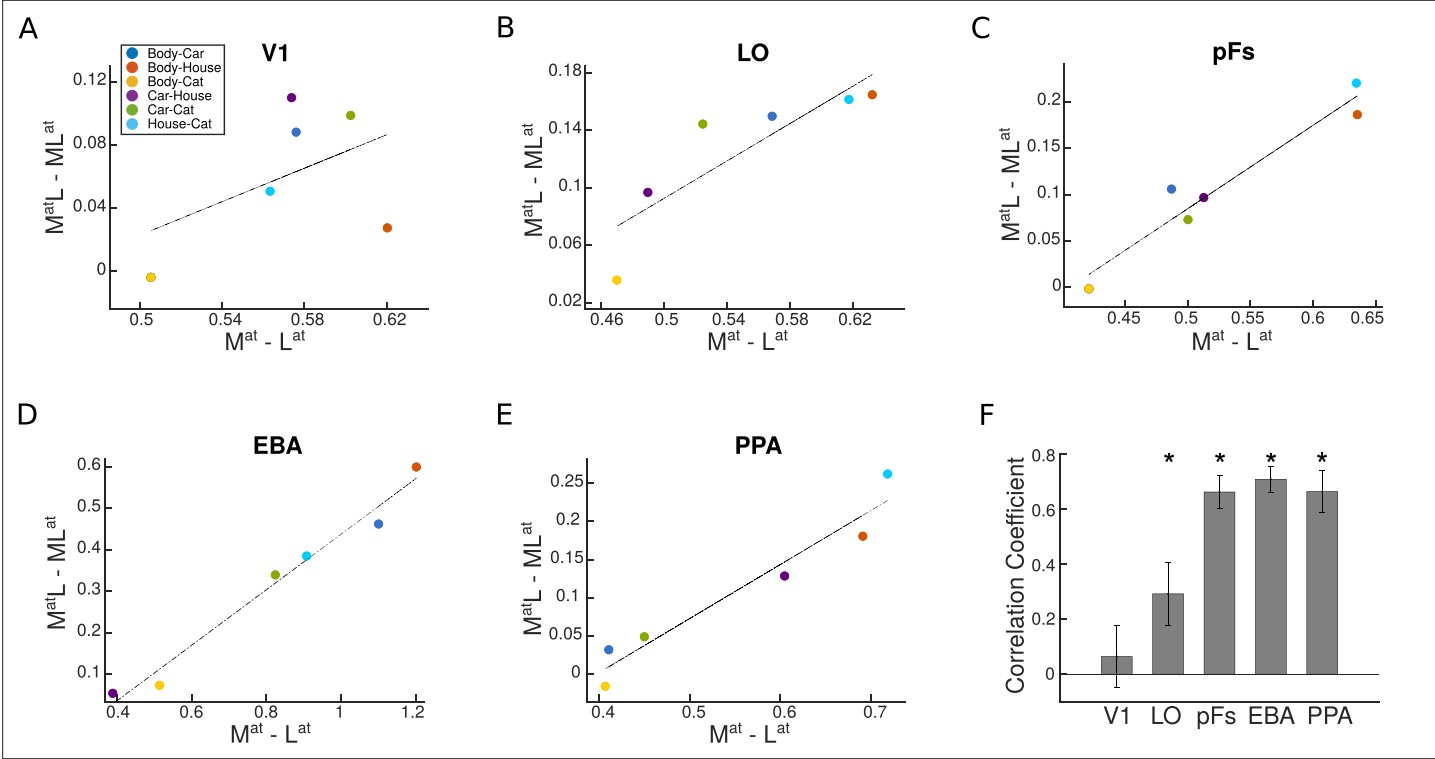

**Figure 5.** Univariate shift versus category distance in each region of interest (ROI). (**A–E**) The value of univariate shift versus category distance. $M^{at}L$ and $ML^{at}$ denote the two paired conditions with attention directed to the more preferred (**M**) or less preferred (**L**) stimulus, respectively. $M^{at}$ and $L^{at}$ represent the isolated conditions, respectively, with the more preferred or the less preferred stimulus presented in isolation. The blue, red, yellow, purple, green, and sky blue circles in each panel represent the values related to the Body-Car, Body-House, Body-Cat, Car-House, Car-Cat, and House-Cat pairs, respectively. The correlation coefficients in each ROI were calculated for single participants and the lines in the average plots are only shown for illustration purposes. (**F**) Correlation coefficient for the correlation between the univariate shift and category distance in each ROI. Asterisks indicate that the correlation coefficients are significantly positive ($p < 0.05$). Error bars represent standard errors of the mean. N=17 human participants.

This indicates that the amount of univariate shift is related to the response difference between the two presented stimuli.

## The multivariate effect of attention decreases for more similar target-distractor pairs

The univariate analysis above considers average response only and thus cannot capture other aspects of response variance. For example, in an object-selective region with diverse selectivity for different objects, the average response to body and house stimuli is close, but the response pattern may be very different since voxels highly responsive to bodies do not show high responses to houses, and vice versa. Thus, we had to consider voxel preferences in the univariate analysis to observe the difference in response between the two categories. Furthermore, although the paired responses can be greater than responses to both isolated conditions (*Figure 4C–D*), there is still the possibility that the univariate shift is limited by the amount of the difference between isolated-condition responses for each category pair.

We complement the univariate approach with a multivariate pattern analysis to assess the relationship between the effect of attention and category distance at the multivariate level. By considering the whole response pattern in an ROI to each stimulus, we can compare the responses to each stimulus without considering voxel preferences. Moreover, using this method we can determine the weight of the response to each isolated stimulus in the total response, and determine the attentional bias related to each category pair.

The multivariate representation of two simultaneously-presented stimuli can be modeled as the linear combination of the representations of the two stimuli presented in isolation (*Reddy et al.*,

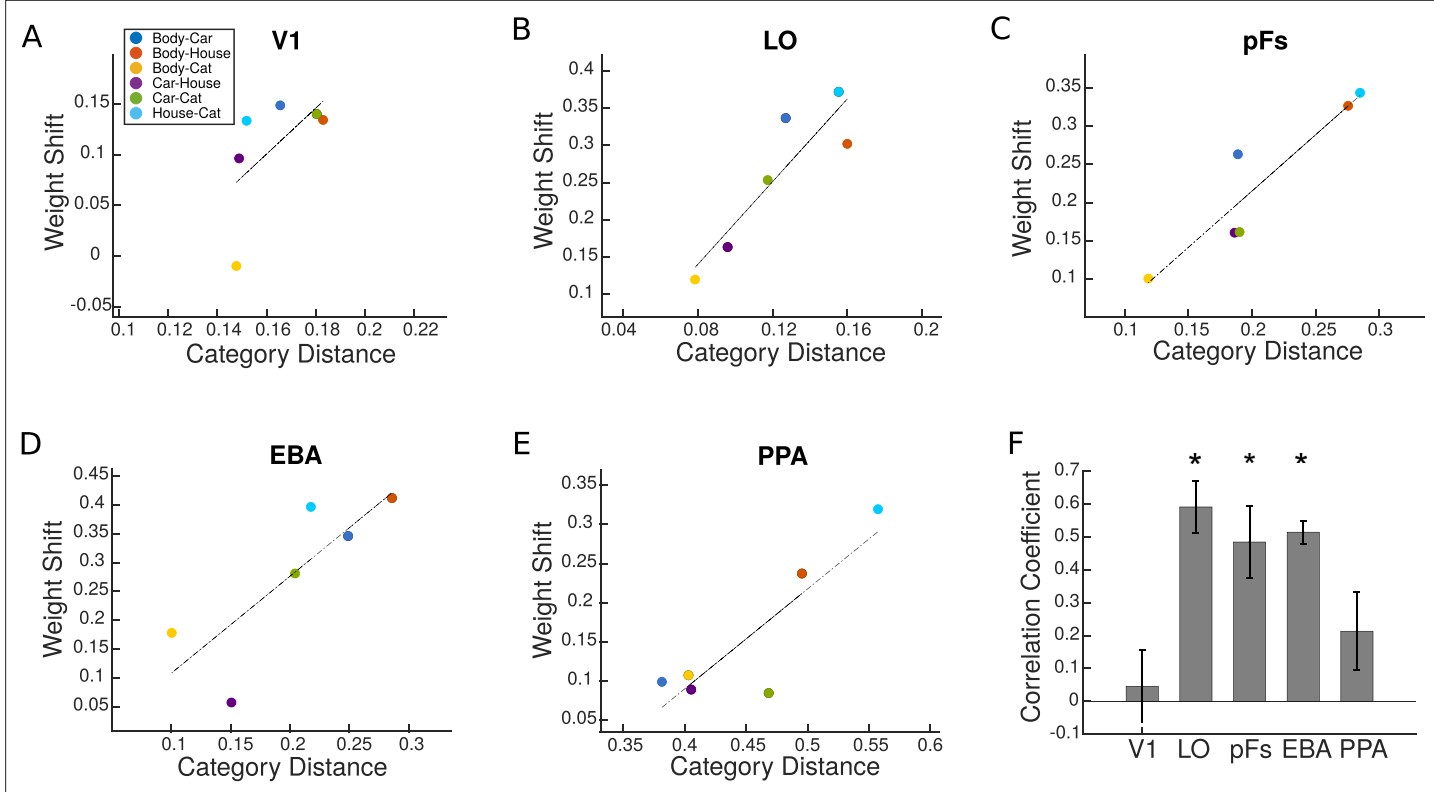

**Figure 6.** Weight shift versus category distance in each region of interest (ROI). (**A–E**) Attentional weight shift versus category distance. The blue, red, yellow, purple, green, and sky blue circles in each panel represent the values related to the Body-Car, Body-House, Body-Cat, Car-House, Car-Cat, and House-Cat pairs, respectively. We calculated the correlation coefficients for single participants and the lines in these average plots are only for illustration purposes. (**F**) Correlation coefficient for the correlation between attentional weight shift and category distance in each ROI. Asterisks indicate that the correlations are significantly positive ($p < 0.05$). Error bars represent standard errors of the mean. N=17 human participants.

The online version of this article includes the following figure supplement(s) for figure 6:

**Figure supplement 1.** Sum of weights in the multivariate analysis for each category pair in each region of interest (ROI), averaged across conditions with attention directed to each of the two categories of a pair and across participants.

*2009*): When one stimulus is attended, the weight of the response to that stimulus increases in the multivariate representation.

Taking this approach, for each category pair (e.g. Body-Car), we considered the multivariate representation of the two paired conditions ($V_{Body^{at}Car}$ and $V_{BodyCar^{at}}$, with $V$ denoting the multivariate response pattern of each condition), and determined the weight of each of the isolated-stimulus responses ($V_{Body^{at}}$ and $V_{Car^{at}}$) in the paired response (*Figure 2B*). We then calculated the difference between the weight of each stimulus when it was the target and when it was the distractor (e.g. for the Body-Car pair, the difference between the weight of $V_{Body^{at}}$ in $V_{Body^{at}Car}$ and $V_{BodyCar^{at}}$).

If attention could perfectly remove the effect of the distractor, the weight of the attended stimulus would equal one and the representation of the pair would be identical to the representation of the isolated target. In this case, the difference between the weight of the stimulus representation when attended and ignored would be a maximal value of one. However, if the distractor is not completely removed, this leads to a weight shift value smaller than one. Thus, the magnitude of the weight shift is an indicator of the efficiency of attention, with greater values indicating a higher efficiency of attention in removing the distractor.

To compare the efficiency of attention across category pairs, we calculated the weight shift for each category pair (*Equation 5*). Similar to the univariate analysis, we took a cross-validation approach and used one-half of the data to calculate the weight shift. Then, to determine whether this multivariate effect of attention was dependent on the similarity between the target and the distractor in their

cortical representation, we calculated the multivariate category distance for each category pair using the second left-out half of the data (*Equation 3*).

As illustrated in *Figure 6A–E*, we observed that the attentional weight shift was not constant for different category pairs, and that weight shift and category distance correlated positively in LO, pFs, and EBA ($ts > 4.4$, $ps < 5 \times 10^{-3}$), marginally significantly in PPA ($t(16) = 1.8$, $p = 0.09$), and not in V1 ($t(16) = 0.42$, $p = 0.68$). Less significant results in PPA might arise from the fact that this region shows no response to body and cat stimuli and little response to car stimuli (see *Appendix 1—table 2*). Therefore, it is not possible to observe the effect of attention for all category pairs in PPA.

We performed the analysis including only voxels that had a significantly positive GLM coefficient across the runs and observed the same results. Moreover, to check whether the effect is robust over more selective thresholds for ROI definition, we redefined the left EBA region with $p < 0.0001$ and $p < 0.00001$ criteria. We observed a similar weight shift effect for both criteria. We also calculated category distance based on the euclidean distance between response patterns of category pairs and observed a similarly positive correlation between the weight shift and the euclidean category distance in all ROIs ($ps < 0.01$, $ts > 2.9$) except V1 ($p = 0.5$, $t = 0.66$). These results are in agreement with our main multivariate results, indicating that the attentional bias towards a stimulus in a pair decreases as the similarity between the two stimuli in neural representation increases.

## Tuning sharpening predicts the dependence of attentional modulation on target-distractor similarity

We observed empirically that attentional enhancement is not constant and content-independent, but rather depends on the response similarity between the target and the distractor. We next asked whether a gain increase or tuning changes predict the observed effect of target-distractor similarity on the attentional bias.

Based on the response gain model, attention increases neural responses by scaling the responses by a constant attention factor (*McAdams and Maunsell, 1999*; *Reynolds and Chelazzi, 2004*). Therefore, the response gain model predicts that attention scales the neurons' tuning function without affecting its shape (*Figure 3A*).

In contrast, the tuning sharpening model proposes that attention enhances the response of each neuron based on its preference for the attended stimulus (*Martinez-Trujillo and Treue, 2004*; *Ling et al., 2009*). Therefore, this model predicts that attention causes a sharpening of the neurons' tuning function, with a sharp increase in the response to optimal stimuli, and no increase in the response to the non-optimal stimuli (*Figure 3B*).

To examine which of these mechanisms could account for the observed results, we simulated the responses of a neural population to isolated and paired stimuli from the four categories of bodies, cars, houses, and cats. Equivalent to the fMRI experiment, we determined neuronal responses to stimuli presented either in isolation or paired with stimuli from another category (*Figure 1B*). We implemented attentional enhancement of the neural responses either using the response gain model (*Equation 6a*), or the tuning sharpening model (*Equation 7a*). We then used the univariate and multivariate analyses equivalent to those used for the fMRI data to determine which model predicts the empirical data.

We created two neural populations: (i) a population with varying selectivity across neurons, representing object-selective regions, in which neurons show different selectivities (similar to LO and pFs), and (ii) a population with similar selectivity across all neurons to represent a region with a strong preference for a specific object category, in which neurons generally show high response to stimuli from that category (similar to EBA and PPA). Then we assessed the univariate shift using the reduction in response when attention shifted from the stronger to the weaker stimulus in a pair (*Equation 2*), and examined its relationship with univariate category distance (*Equation 1*).

We found that the response gain model predicted no relationship between univariate shift and category distance in either population (*Figure 7A–B*). In contrast, the tuning sharpening model predicted a positive correlation between univariate shift and category distance in both neural populations (*Figure 7C–D*). Thus, the tuning sharpening model provides a better prediction of the empirical data compared to the response gain model.

Next, for the multivariate analysis, we assessed the attentional weight shift for each category pair as attention shifted from one stimulus to the other (*Equation 5*), and examined its relationship with

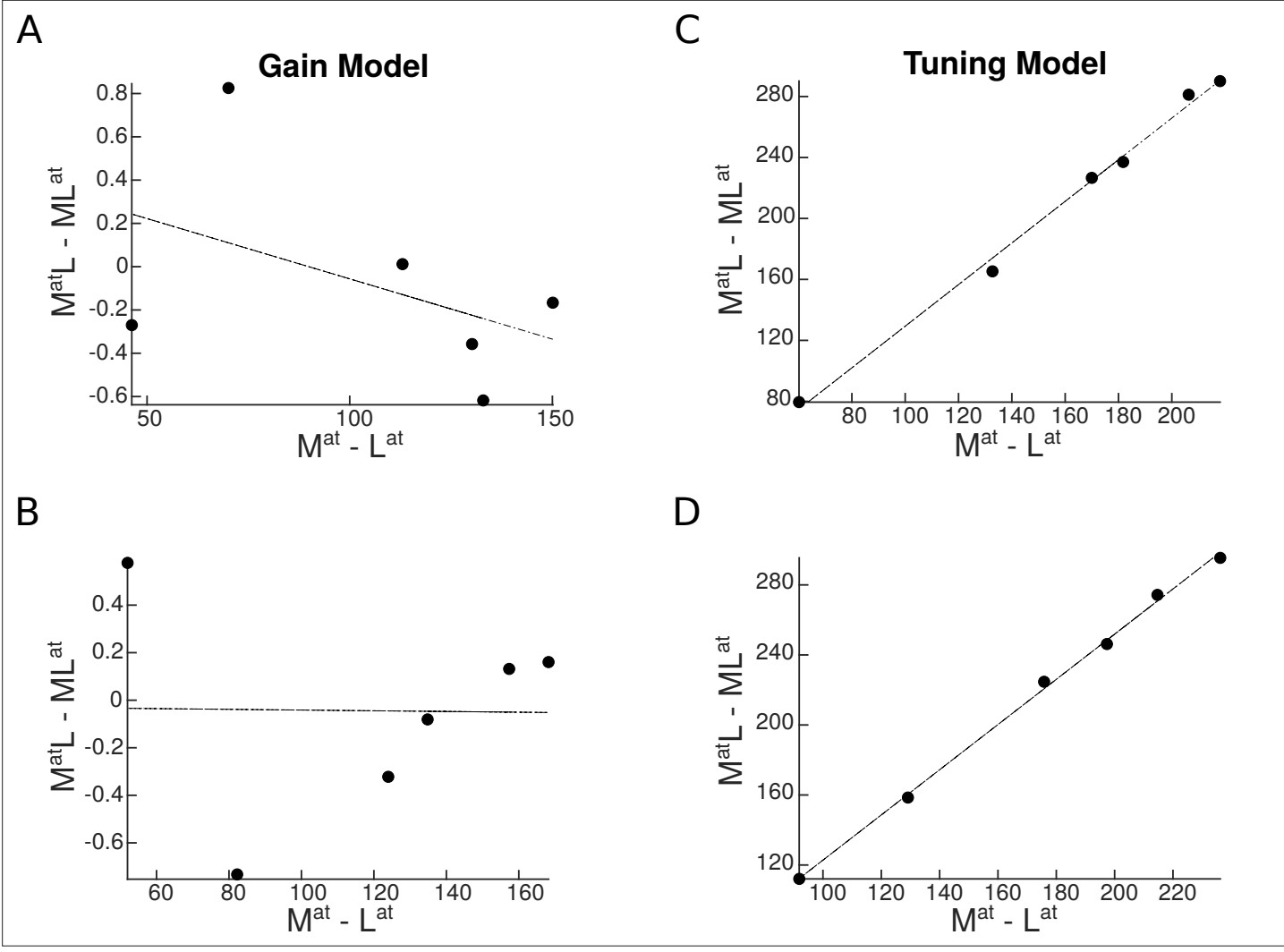

**Figure 7.** Univariate shift as a function of category distance, as predicted by the two attentional mechanisms. $M^{at}L$ and $ML^{at}$ denote the two paired conditions with attention directed to the more preferred ($M$) or the less preferred ($L$) stimulus, respectively. $M^{at}$ and $L^{at}$ represent the isolated conditions, respectively with the $M$ or the $L$ stimulus presented in isolation. Top panels represent predictions in a region with a strong preference for a specific category, and bottom panels illustrate predictions in an object-selective region. Each circle represents a pair of categories. (**A–B**) Predicted univariate shift based on the response gain model in a region with a strong preference for a specific category (**A**) and in an object-selective region (**B**). (**C–D**) Predicted univariate shift based on the tuning model in a region with strong preference for a specific category (**C**) and in an object-selective region (**D**).

the multivariate category distance (*Equation 3*). Here, too, we find that the response gain model predicted no relationship between attentional weight shift and category distance (*Figure 8A–B*). In contrast, the tuning sharpening model predicted a positive relationship between weight shift and category distance for both neural populations (*Figure 8C–D*), providing further evidence for tuning sharpening as the underlying mechanism for attentional enhancement.

We also tested a third model based on a labeled line mechanism for attentional enhancement (see Appendix 1). The labeled line model posits that attention to a stimulus enhances the neural response only when the attended stimulus is the neuron's preferred stimulus. Therefore, this model is a special case of change in the neurons' tuning curve (*Appendix 1—figure 1*). Although the labeled line model could predict the positive correlation between the univariate shift and category distance in a region with high selectivity for a certain category, it failed to predict the results in other cases (*Appendix 1—figure 2*).

In sum, the tuning model predicts the empirically-observed effect of target-distractor similarity on attentional effects both at the univariate and at the multivariate level, while the response gain model does not.

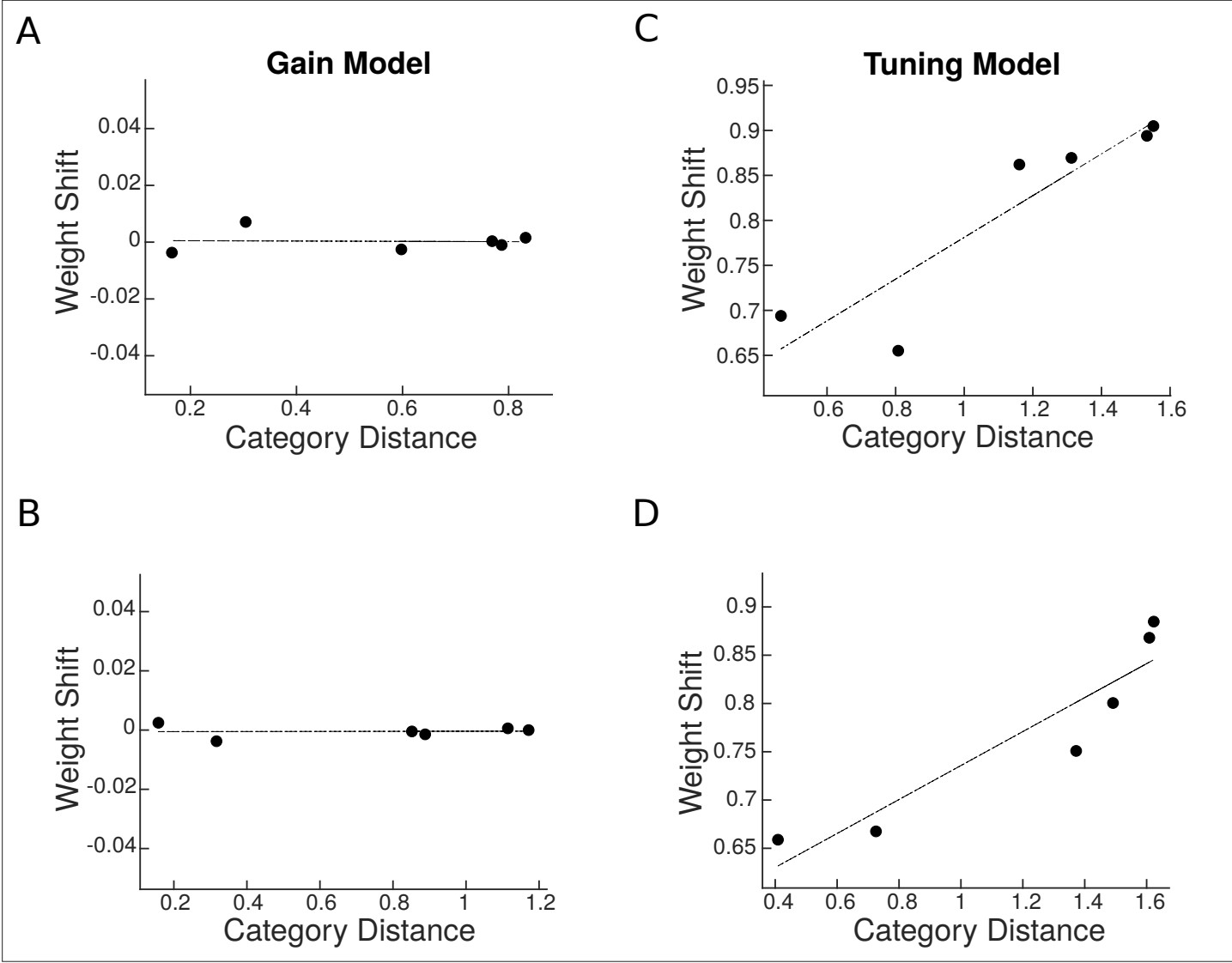

**Figure 8.** Predicted weight shift as a function of category distance. Weight shift for each pair is calculated using *Equation 5*. Category distance represents the difference in multi-voxel representation between responses to the two isolated stimuli, calculated by *Equation 3*. Top panels are related to predictions in a region with a strong preference for a specific category and the bottom panels illustrate predictions in an object-selective region. (**A–B**) Weight shift predicted by the response gain model in a region with a strong preference for a specific category (**A**) and in an object-selective region (**B**). (**C–D**) Weight shift predicted by the tuning model in a region with strong preference for a specific category (**C**) and in an object-selective region (**D**).

## Discussion

Visual stimuli compete for resources in the brain. The biased competition model posits that attention to a stimulus biases this competition in favor of the attended stimulus (*Moran and Desimone, 1985*; *Desimone and Duncan, 1995*; *Reynolds et al., 1999*). Here, we examined the change in this attentional bias by systematically varying the target and distractors. Using fMRI, we showed that rather than being a constant top-down bias, attentional enhancement depends on the similarity between the target and the distractor in their cortical representation, both at the univariate level and at the multivariate level. Using simulations, we arbitrated between the response gain model and the tuning sharpening model as mechanisms of attention for the observed effect, and showed that the empirical results were explained by the latter and not the former.

## Effect of target-distractor similarity on the attentional bias

Using stimuli from four object categories, our study reveals the neural basis of the attentional effect graded by target-distractor similarity in the human brain both at the univariate level and at the multivariate level. This finding has two important implications:

First, our results show that in the competition between multiple stimuli, the attentional bias is not constant. Previous studies have shown attentional modulation in the human brain as an average value without considering its variance for different pairings of targets and distractors (*Reddy et al., 2009*). These previous accounts of attention cannot explain the variance in performance for the same number of stimuli from different categories. Assessing the role of stimulus content in the bias caused by attention, we confirm that attention enhances the response related to the target. We refine our understanding by showing that however, the attentional bias offers less advantage for a more similar target-distractor pair.

Second, this finding provides direct neural evidence for the adverse effects of target-distractor similarity on performance, as previously reported in behavioral studies (*Cohen et al., 2014*; *Cohen et al., 2017*). While behavioral data have suggested that this effect is due to limitations in processing, no investigation has been made to determine the underlying reason or find a mechanistic explanation. Our results demonstrate that this reduction in performance is because the representation of the target (relative to the distractor) is less effectively enhanced by attention when the target becomes more similar to the distractor.

We observed a significant univariate shift in higher-level regions of the occipito-temporal cortex, but not in V1. Evidence on the effect of attention on V1 responses is divergent, with some previous neuroimaging studies showing a significant effect of attention on neural responses (*Somers et al., 1999*; *Gandhi et al., 1999*), while others report no significant effect of attention (*Corbetta et al., 1990*; *Thorat and Peelen, 2022*; *Doostani et al., 2023*). We believe that this apparent discrepancy results from the form of attention under study. Here, we study object-based attention with a superimposed design that excludes response modulation by space-based attention. Previous reports of significant attentional modulation in V1 include studies of space-based attention with stimuli presented at different locations (*Somers et al., 1999*; *Gandhi et al., 1999*). Considering the high reliance of V1 responses to location, the effect of attention is less pronounced when the two stimuli are presented at the same location, as is the case in the present study.

Although examples of superimposed cluttered stimuli are not very common in everyday life, they still do occur in certain situations, for example, reading text on the cellphone screen in the presence of reflection and glare on the screen or looking at the street through a patterned window. Such instances recruit object-based attention which was the aim of this study, whereas in more common cases in which attended and unattended objects occupy different locations in space, both space-based and object-based attention may work together to resolve the competition between different stimuli. Here, we chose to move away from usual everyday scenarios to study the effect of object-based attention in isolation. Future studies can reveal the effect of target-distractor similarity, i.e., proximity in space, on space-based attention and how the effects caused by object-based and space-based attention interact.

Please note that we used a blocked design in which the target and distractor categories could be predicted across each block. While it is possible that the current design has led to an enhancement of the observed effect, previous behavioral data *Cohen et al., 2014*; *Xu and Vaziri-Pashkam, 2019* have reported the same effect in experiments in which the distractor was not predictable. To study the effect of predictability on fMRI responses, however, an event-related design is more appropriate, an interesting venue for future fMRI studies.

## A model for object-based attentional enhancement

Using a simulation approach, we provide a mechanistic explanation for the observed graded attentional effect. Our modeling results have two implications:

First, we demonstrate that tuning sharpening, but not response gain, predicts the observed reduction in the effect of attention for more similar target-distractor pairs both at the univariate and at the multivariate level. Previous research has shown that a change in the tuning function improves attentional selection at high external noise levels (*Ling et al., 2009*). Our results indicate that a change in tuning function could also lead to behavioral disadvantage in an environment where the target is not

very different from the surrounding items. When attention is directed towards the target, the response to non-target objects that are more similar to the target is also enhanced, albeit to a lesser amount, leading to an overall weaker effect of attention for a more similar target-distractor pair.

Second, providing evidence from the human brain in favor of tuning sharpening, we suggest tuning sharpening as the underlying mechanism in the domain of object-based attention. By comparing the response gain model and the tuning sharpening model directly in a single study, we provide strong evidence that arbitrates between the theories. The effects of attention have generally been explained by attention acting through increasing the contrast or response gain, especially for space-based attention (*McAdams and Maunsell, 1999*; *Reynolds and Chelazzi, 2004*; *Fox et al., 2023*). However, a simple increase in gain cannot explain all reported effects of attention, and a change in the shape of the tuning curves has been observed during visual search (*Çukur et al., 2013*), and feature-based attention (*Martinez-Trujillo and Treue, 2004*; *David et al., 2008*; *Ling et al., 2009*).

While tuning curves are commonly used for feature dimensions such as stimulus orientation or motion direction, here, we used the term to describe the variation in a neuron's response to different object stimuli. With a finite set of object categories, as in the current study, the neural response in object space is discrete, rather than a continuous curve illustrated for features such as stimulus orientation. The neuron might have to tune for a particular feature such as curvature or spikiness (*Bao et al., 2020*) that is present to different degrees in our object stimuli in a continuous way, but we are not measuring this directly. Nevertheless, since more preferred and less preferred features (objects in this case) can still be defined, we illustrate the neural response using a hypothetical curve in object space. As such, here, tuning sharpening refers to the fact that the response to the more preferred object categories has been enhanced while the response to the less preferred stimulus categories is suppressed.

It is important to note that our speculation on the role of tuning sharpening in object-based attention is based on simulations and not neural data. To ascertain tuning sharpening as the underlying mechanism for object-based attention, intracranial recordings from the human brain are needed.

## Conclusion

In sum, our results unravel the cortical basis by which target-distractor similarity affects attentional modulation, and indicate tuning sharpening as the underlying mechanism for response enhancement during object-based attention.

## Acknowledgements

We thank Sajad Aghapour for the helpful discussions. We thank Kiarash Farahmandrad for his help with the graphical illustration of the vector plot. Maryam Vaziri-Pashkam was supported by NIH Intramural Research Program ZIA-MH002035.

## Additional information

### Funding

| Funder | Grant reference number | Author |
|---|---|---|
| National Institutes of Health | ZIA-MH002035 | Maryam Vaziri-Pashkam |

The funders had no role in study design, data collection and interpretation, or the decision to submit the work for publication.

### Author contributions

Narges Doostani, Conceptualization, Data curation, Formal analysis, Investigation, Visualization, Methodology, Writing - original draft, Writing – review and editing; Gholam-Ali Hossein-Zadeh, Resources, Supervision, Validation, Methodology, Writing – review and editing; Radoslaw M Cichy, Supervision, Validation, Writing – review and editing; Maryam Vaziri-Pashkam, Conceptualization, Supervision, Funding acquisition, Validation, Methodology, Project administration, Writing – review and editing

## Author ORCIDs
Narges Doostani ![ORCID] https://orcid.org/0000-0001-5775-6595
Radoslaw M Cichy ![ORCID] https://orcid.org/0000-0003-4190-6071
Maryam Vaziri-Pashkam ![ORCID] https://orcid.org/0000-0003-1830-2501

## Ethics
Participants gave written consent and received payment for their participation in the experiment. Data collection was approved by the Ethics Committee of the Institute for Research in Fundamental Sciences, Tehran. Reference number for the ethical approval: 98/60/2184.

Reviewer #1 (Public review): https://doi.org/10.7554/eLife.89836.3.sa1
Author response https://doi.org/10.7554/eLife.89836.3.sa2

---

# Additional files

## Supplementary files
• MDAR checklist

## Data availability
fMRI data have been deposited in OSF under https://doi.org/10.17605/OSF.IO/2QTF6.

The following dataset was generated:

| Author(s) | Year | Dataset title | Dataset URL | Database and Identifier |
|---|---|---|---|---|
| Doostani N, Vaziri-Pashkam M | 2023 | Similarity | https://doi.org/10.17605/OSF.IO/2QTF6 | Open Science Framework, 10.17605/OSF.IO/2QTF6 |

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

## Appendix 1

### The labeled line model

We also simulated a special case of change in the tuning curve called the labeled line model. Based on this model, attention to a certain stimulus enhances the neural response only if the neuron is specifically labeled for that stimulus (*Appendix 1—figure 1*). For instance, attention to body stimuli causes an enhancement in the response of body neurons, but no enhancement in the response of car neurons which might respond to body stimuli to a lesser level. We implemented this mechanism using *Equation 8a*: for a body-selective neuron:

$$R_{Body^{at}} = \beta \cdot R_{Body} \tag{8a}$$

$$R_{Car^{at}} = R_{Car} \tag{8b}$$

For a car-selective neuron:

$$R_{Body^{at}} = R_{Body} \tag{8c}$$

$$R_{Car^{at}} = \beta \cdot R_{Car} \tag{8d}$$

For a house-selective neuron:

$$R_{Body^{at}} = R_{Body} \tag{8e}$$

$$R_{Car^{at}} = R_{Car} \tag{8f}$$

Then, using the labeled line mechanism for attentional enhancement, we simulated the response of two neural populations in the 16 task conditions (see Materials and methods and Results). Performing the univariate analysis on the simulated responses, we assessed the univariate shift for all category pairs. The labeled line model predicted a positive correlation between the univariate shift and category distance in the population with a strong preference for a certain category, while it predicted no relationship between univariate shift and category distance in the object-selective population (*Appendix 1—figure 2A, B*).

In the multivariate analysis, the labeled line model predicted no relationship for the neural population with a strong preference for a certain category (*Appendix 1—figure 2C*), while it predicted a negative correlation between weight shift and category distance in the object-selective population (*Appendix 1—figure 2D*).

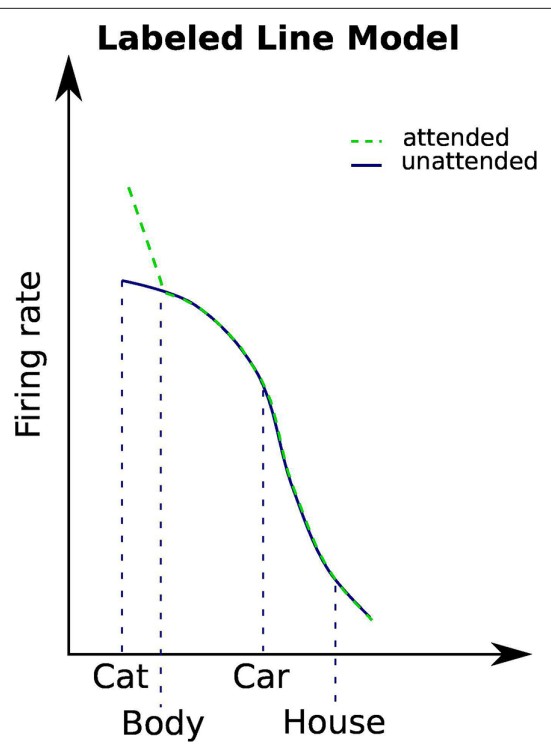

**Appendix 1—figure 1.** Based on the labeled line model, attention enhances the response of a neuron only when the attended stimulus is the neuron's preferred stimulus. We illustrate the models here for the example of a neuron with high selectivity for cat stimuli. Solid curves denote the response to unattended stimuli and dashed curves denote the response to attended stimuli.

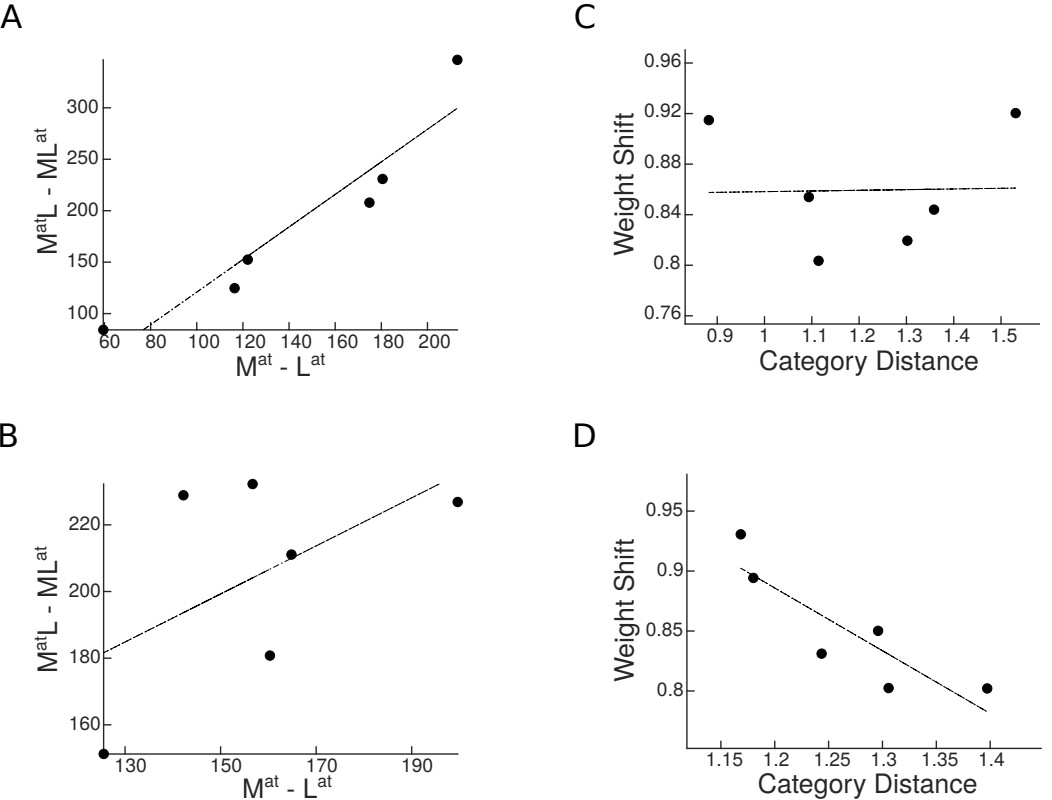

**Appendix 1—figure 2.** Simulation results for the labeled line model. (**A–B**) Predicted univariate shift based on the labeled line model in a region with a strong preference for a specific category (**A**) and in an object-selective region (**B**). (**C–D**) Weight shift predicted by the labeled line model in a region with a strong preference for a specific category (**C**) and in an object-selective region (**D**).

**Appendix 1—table 1.** Percentage of voxels in each regions of interest (ROI) most responsive to each of the four stimulus categories, averaged across participants.

| Preferred Category | V1 | LO | pFs | EBA | PPA |
|---|---|---|---|---|---|
| Body | 21% | 31% | 21% | 70% | 10% |
| Car | 19% | 18% | 17% | 4% | 12% |
| House | 33% | 32% | 40% | 3% | 68% |
| Cat | 27% | 19% | 22% | 23% | 10% |

**Appendix 1—table 2.** Average voxel response (general linear model, GLM coefficients) to each category in each region of interest (ROI), averaged across participants.

| Stimulus Category | V1 | LO | pFs | EBA | PPA |
|---|---|---|---|---|---|
| Body | 0.28 | 1.59 | 0.88 | 1.62 | 0.001 |
| Car | 0.28 | 1.43 | 0.88 | 0.60 | 0.12 |
| House | 0.47 | 1.51 | 1.03 | 0.48 | 0.55 |
| Cat | 0.35 | 1.49 | 0.89 | 1.32 | –0.02 |

**Appendix 1—table 3.** Single-subject correlation coefficients for the correlation between the univariate shift and category distance in all regions of interest (ROIs).

| V1 | LO | pFs | EBA | PPA |
|---|---|---|---|---|
| –0.60 | 0.67 | –0.073 | 0.72 | 0.18 |
| 0.16 | 0.38 | 0.015 | 0.64 | 0.46 |
| –0.53 | 0.52 | 0.89 | 0.60 | 0.42 |
| –0.062 | 0.78 | 0.93 | 0.87 | 0.69 |
| –0.098 | 0.072 | 1.0 | 0.72 | 0.96 |
| 0.63 | 0.93 | 0.88 | 0.94 | 0.58 |
| –0.31 | 0.52 | –0.11 | 0.88 | 0.89 |
| 0.29 | 0.18 | 0.62 | 0.67 | 0.63 |
| –0.49 | 0.094 | 0.63 | 0.81 | 0.80 |
| 0.21 | 0.57 | 0.31 | 0.81 | 0.87 |
| –0.015 | –0.57 | 0.79 | 0.35 | 0.69 |
| 0.35 | 0.79 | 0.90 | 0.90 | 0.88 |
| –0.47 | 0.058 | –0.050 | 0.93 | –0.24 |
| –0.047 | –0.21 | 0.91 | 0.28 | 0.86 |
| 0.66 | 0.70 | 0.91 | 0.64 | 0.45 |
| –0.74 | 0.54 | 0.24 | 0.60 | –0.31 |
| –0.095 | 0.73 | 0.89 | 0.64 | 0.95 |

**Appendix 1—table 4.** Weights of each stimulus for each category pair in each region of interest (ROI), averaged across participants.

| Pair | Attended Stimulus | V1 | LO | pFs | EBA | PPA |
|---|---|---|---|---|---|---|
| Body-Car | Body | $0.54 \times R_{B}at + 0.53 \times R_{Cr}at$ | $0.79 \times R_{B}at + 0.31 \times R_{Cr}at$ | $0.68 \times R_{Ba_1} + 0.42 \times R_{Cr}at$ | $0.93 \times R_{B}at + 0.15 \times R_{Cr}at$ | $0.41 \times R_{B}at + 0.42 \times R_{Cr}at$ |
| Body-Car | Car | $0.40 \times R_{B}at + 0.70 \times R_{Cr}at$ | $0.43 \times R_{B}at + 0.68 \times R_{Cr}at$ | $0.39 \times R_{B}at + 0.70 \times R_{Cr}at$ | $0.53 \times R_{B}at + 0.46 \times R_{Cr}at$ | $0.35 \times R_{B}at + 0.54 \times R_{Cr}at$ |
| Body-House | Body | $0.43 \times R_{B}at + 0.65 \times R_{H}at$ | $0.65 \times R_{B}at + 0.45 \times R_{H}at$ | $0.65 \times R_{B}at + 0.47 \times R_{H}at$ | $0.84 \times R_{B}at + 0.21 \times R_{H}at$ | $0.42 \times R_{B}at + 0.61 \times R_{H}at$ |
| Body-House | House | $0.27 \times R_{B}at + 0.78 \times R_{H}at$ | $0.3 \times R_{B}at + 0.74 \times R_{H}at$ | $0.27 \times R_{B}at + 0.81 \times R_{H}at$ | $0.37 \times R_{B}at + 0.57 \times R_{H}at$ | $0.18 \times R_{B}at + 0.86 \times R_{H}at$ |
| Body-Cat | Body | $0.63 \times R_{B}at + 0.55 \times R_{Ct}at$ | $0.86 \times R_{B}at + 0.35 \times R_{Ct}at$ | $0.81 \times R_{B}at + 0.45 \times R_{Ct}at$ | $0.90 \times R_{B}at + 0.30 \times R_{Ct}at$ | $0.61 \times R_{B}at + 0.32 \times R_{Ct}at$ |
| Body-Cat | Cat | $0.59 \times R_{B}at + 0.55 \times R_{Ct}at$ | $0.69 \times R_{B}at + 0.48 \times R_{Ct}at$ | $0.67 \times R_{B}at + 0.55 \times R_{Ct}at$ | $0.67 \times R_{B}at + 0.48 \times R_{Ct}at$ | $0.55 \times R_{B}at + 0.37 \times R_{Ct}at$ |
| Car-House | Car | $0.42 \times R_{Cr}at + 0.65 \times R_{H}at$ | $0.44 \times R_{Cr}at + 0.70 \times R_{H}at$ | $0.55 \times R_{Cr}at + 0.60 \times R_{H}at$ | $0.56 \times R_{Cr}at + 0.59 \times R_{H}at$ | $0.35 \times R_{Cr}at + 0.72 \times R_{H}at$ |
| Car-House | House | $0.31 \times R_{Cr}at + 0.73 \times R_{H}at$ | $0.25 \times R_{Cr}at + 0.85 \times R_{H}at$ | $0.36 \times R_{Cr}at + 0.78 \times R_{H}at$ | $0.46 \times R_{Cr}at + 0.65 \times R_{H}at$ | $0.27 \times R_{Cr}at + 0.93 \times R_{H}at$ |
| Car-Cat | Car | $0.59 \times R_{Cr}at + 0.52 \times R_{Ct}at$ | $0.68 \times R_{Cr}at + 0.44 \times R_{Ct}at$ | $0.74 \times R_{Cr}at + 0.40 \times R_{Ct}at$ | $0.47 \times R_{Cr}at + 0.54 \times R_{Ct}at$ | $0.53 \times R_{Cr}at + 0.40 \times R_{Ct}at$ |
| Car-Cat | Cat | $0.45 \times R_{Cr}at + 0.70 \times R_{Ct}at$ | $0.41 \times R_{Cr}at + 0.75 \times R_{Ct}at$ | $0.55 \times R_{Cr}at + 0.60 \times R_{Ct}at$ | $0.23 \times R_{Cr}at + 0.92 \times R_{Ct}at$ | $0.46 \times R_{Cr}at + 0.46 \times R_{Ct}at$ |
| House-Cat | House | $0.76 \times R_{H}at + 0.32 \times R_{Ct}at$ | $0.87 \times R_{H}at + 0.24 \times R_{Ct}at$ | $0.91 \times R_{H}at + 0.23 \times R_{Ct}at$ | $0.56 \times R_{H}at + 0.44 \times R_{Ct}at$ | $0.99 \times R_{H}at + 0.11 \times R_{Ct}at$ |
| House-Cat | Cat | $0.61 \times R_{H}at + 0.47 \times R_{Ct}at$ | $0.47 \times R_{H}at + 0.65 \times R_{Ct}at$ | $0.53 \times R_{H}at + 0.62 \times R_{Ct}at$ | $0.17 \times R_{H}at + 0.90 \times R_{Ct}at$ | $0.57 \times R_{H}at + 0.38 \times R_{Ct}at$ |

**Appendix 1—table 5.** Single-subject correlation coefficients for the correlation between weight shift and category distance in all regions of interests (ROIs).

| V1 | LO | pFs | EBA | PPA |
|---|---|---|---|---|
| –0.36 | 0.93 | –0.25 | 0.46 | 0.06 |
| 0.31 | –0.06 | 0.16 | 0.82 | –0.25 |

*Appendix 1—table 5 Continued on next page*

*Appendix 1—table 5 Continued*

| V1 | LO | pFs | EBA | PPA |
|---|---|---|---|---|
| –0.29 | 0.35 | 0.80 | 0.48 | –0.42 |
| –0.038 | 0.70 | 0.75 | 0.57 | 0.38 |
| 0.72 | 0.60 | 0.90 | 0.58 | –0.77 |
| 0.53 | 0.69 | 0.76 | 0.51 | 0.59 |
| –0.32 | 0.90 | –0.25 | 0.29 | 0.065 |
| 0.33 | 0.22 | –0.36 | 0.24 | –0.48 |
| 0.52 | 0.31 | 0.63 | 0.54 | 0.91 |
| 0.019 | 0.96 | 0.80 | 0.61 | 0.71 |
| –0.11 | 0.87 | 0.89 | 0.69 | 0.51 |
| –0.27 | 0.94 | 0.81 | 0.63 | 0.50 |
| –0.28 | 0.28 | –0.11 | 0.49 | 0.59 |
| 0.42 | 0.11 | 0.61 | 0.42 | 0.59 |
| 0.67 | 0.77 | 0.78 | 0.36 | 0.36 |
| –0.96 | 0.78 | 0.48 | 0.43 | –0.24 |
| –0.12 | 0.69 | 0.82 | 0.63 | 0.56 |

