## [Editor Report · eLife Assessment]

This **valuable** study has the potential to shed mechanistic light on how attention mechanisms that influence competition between multiple visual stimuli are modulated by the relative neural similarity of these stimuli. The study provides **convincing** data that will also be used for future modeling efforts. The study will be of interest to researchers working on the neural basis of visual attention.

---

## [Referee Report · Reviewer #1 (Public review)]

Summary:

The authors report an fMRI investigation of the neural mechanisms by which selective attention allows capacity-limited perceptual systems to preferentially represent task-relevant visual stimuli. Specifically, they examine competitive interactions between two simultaneously-presented items from different categories, to reveal how task-directed attention to one of them modulates the activity of brain regions that respond to both. The specific hypothesis is that attention will bias responses to be more like those elicited by the relevant object presented on its own, and further that this modulation will be stronger for more dissimilar stimulus pairs. This pattern was confirmed in univariate analyses that measured the mass response of a priori regions of interest, as well as multivariate analyses that considered the patterns of evoked activity within the same regions. The authors follow these neuroimaging results with a simulation study that favours a "tuning" mechanism of attention (enhanced responses to highly effective stimuli, and suppression for ineffective stimuli) to explain this pattern.

Strengths:

The manuscript clearly articulates a core issue in the cognitive neuroscience of attention, namely the need to understand how limited perceptual systems cope with complex environments in the service of the observer's goals. The use of a priori regions of interest (and a control region), and the inclusion of both univariate and multivariate analyses as well as a simple model, are further strengths. The authors carefully derive clear indices of attentional effects (for both univariate and multivariate analyses) which makes explication of their findings easy to follow.

Weaknesses:

Direct estimation of baseline responses may have improved the validity of the modelling. The presentation of transparently overlapping items has some methodological advantages, but somewhat limits the ecological validity of connections to real-world visual "clutter".

---

## [Author Response]

The following is the authors’ response to the original reviews.

**Public Reviews:**

**Reviewer #1 (Public Review):**
Summary:The authors report an fMRI investigation of the neural mechanisms by which selective attention allows capacity-limited perceptual systems to preferentially represent task-relevant visual stimuli. Specifically, they examine competitive interactions between two simultaneously-presented items from different categories, to reveal how task-directed attention to one of them modulates the activity of brain regions that respond to both. The specific hypothesis is that attention will bias responses to be more like those elicited by the relevant object presented on its own, and further that this modulation will be stronger for more dissimilar stimulus pairs. This pattern was confirmed in univariate analyses that measured the mass response of a priori regions of interest, as well as multivariate analyses that considered the patterns of evoked activity within the same regions. The authors follow these neuroimaging results with a simulation study that favours a "tuning" mechanism of attention (enhanced responses to highly effective stimuli, and suppression for ineffective stimuli) to explain this pattern.Strengths:The manuscript clearly articulates a core issue in the cognitive neuroscience of attention, namely the need to understand how limited perceptual systems cope with complex environments in the service of the observer's goals. The use of a priori regions of interest, and the inclusion of both univariate and multivariate analyses as well as a simple model, are further strengths. The authors carefully derive clear indices of attentional effects (for both univariate and multivariate analyses) which makes explication of their findings easy to follow.Weaknesses:There are some relatively minor weaknesses in presentation, where the motivation behind some of the procedural decisions could be clearer. There are some apparently paradoxical findings reported -- namely, cases in which the univariate response to pairs of stimuli is greater than to the preferred stimulus alone -- that are not addressed. It is possible that some of the main findings may be attributable to range effects: notwithstanding the paradox just noted, it seems that a floor effect should minimise the range of possible attentional modulation of the responses to two highly similar stimuli. One possible limitation of the modelled results is that they do not reveal any attentional modulation at all under the assumptions of the gain model, for any pair of conditions, implying that as implemented the model may not be correctly capturing the assumptions of that hypothesis.

We thank the reviewer for the constructive comments. In response, in the current version of the manuscript we have improved the presentation. We further discuss how the response in paired conditions is in some cases higher than the response to the preferred stimulus in this letter. For this, we provide a vector illustration, and a supplementary figure of the sum of weights to show that the weights of isolated-stimulus responses for each category pair are not bound to the similarity of the two isolated responses.

Regarding the simulation results, we have clarified that the univariate effect of attention is not the attentional modulation itself, but the change in the amount of attentional modulation in the two paired conditions. We provide an explanation for this in this letter below, and have changed the term “attentional modulation” to “univariate shift” in the manuscript to avoid the confusion.

**Reviewer #2 (Public Review):**
Summary:In an fMRI study requiring participants to attend to one or another object category, either when the object was presented in isolation or with another object superimposed, the authors compared measured univariate and multivariate activation from object-selective and early visual cortex to predictions derived from response gain and tuning sharpening models. They observed a consistent result across higher-level visual cortex that more-divergent responses to isolated stimuli from category pairs predicted a greater modulation by attention when attending to a single stimulus from the category pair presented simultaneously, and argue via simulations that this must be explained by tuning sharpening for object categories.Strengths:- Interesting experiment design & approach - testing how category similarity impacts neural modulations induced by attention is an important question, and the experimental approach is principled and clever.- Examination of both univariate and multivariate signals is an important analysis strategy.- The acquired dataset will be useful for future modeling studies.Weaknesses:- The experimental design does not allow for a neutral 'baseline' estimate of neural responses to stimulus categories absent attention (e.g., attend fixation), nor of the combination of the stimulus categories. This seems critical for interpreting results (e.g., how should readers understand univariate results like that plotted in Fig. 4C-D, where the univariate response is greater for 2 stimuli than one, but the analyses are based on a shift between each extreme activation level?).

We are happy to clarify our research rationale. We aimed to compare responses in paired conditions when the stimuli were kept constant while varying the attentional target. After we showed that the change in the attentional target resulted in a response change , we compared the amount of this response change to different stimulus category pairs to investigate the effect of representation similarity between the target and the distractor on the response modulation caused by attentional shift. While an estimate of the neural responses in the absence of attention might be useful for other modeling studies, it would not provide us with more information than the current data to answer the question of this study.

Regarding the univariate results in Fig. 4C-D (and other equivalent ROI results in the revised version) and our analyses, we did not impose any limit on the estimated weights of the two isolated responses in the paired response and thus the sum of the two weights could be any number. We however see that the naming of “weighted average”, which implies a sum of weights being capped at one, has been misleading . We have now changed the name of this model to “linear combination” to avoid confusion

Previous studies (Reddy et al., 2009, Doostani et al., 2023) using a similar approach have shown a related results pattern: the response to multiple stimuli is higher than the average, but lower than the sum of the isolated responses, which is exactly what our results suggest. We have added discussion on this topic in the Results section in lines 409-413 for clarification:

“Note that the response in paired conditions can be higher or lower than the response to the isolated more preferred stimulus (condition Mat), depending on the voxel response to the two presented stimuli, as previously reported (Doostani et al. 2023). This is consistent with previous studies reporting the response to multiple stimuli to be higher than the average, but lower than the sum of the response to isolated stimuli (Reddy et al. 2009).”

We are not sure what the reviewer means by “each extreme activation level”. Our analyses are based on all four conditions. The two isolated conditions are used to calculate the distance measures and the two paired conditions are used for calculating the shift index. Please note that either the isolated or the paired conditions could show the highest response and we seeboth cases in our data. For example, as shown in Figure 4A in EBA, the isolated Body condition and the paired BodyatCar condition show the highest activation levels for the Body-Car pair, whereas in Figure 4C, the two paired conditions (BodyatCat and BodyCatat) elicit the highest response.

- Related, simulations assume there exists some non-attended baseline state of each individual object representation, yet this isn't measured, and the way it's inferred to drive the simulations isn't clearly described.

We agree that the simulations assume a non-attended baseline state, and that we did not measure that state empirically. We needed this non-attended response in the simulations to test which attention mechanism led to the observed results. Thus, we generated the non-attended response using the data reported in previous neural studies of object recognition and attention in the visual cortex (Ni et al., 2012, Bao and Tsao, 2018). Note that the simulations are checking for the profile of the modulations based on category distance. Thus, they do not need to exactly match the real isolated responses in order to show the effect of gain and tuning shift on the results. We include the clarification and the range of neural responses and attention parameters used in the simulations in the revised manuscript in lines 327-333:

“To examine which attentional mechanism leads to the effects observed in the empirical data, we generated the neural response to unattended object stimuli as a baseline response in the absence of attention, using the data reported by neural studies of object recognition in the visual cortex (Ni et al., 2012, Bao and Tsao, 2018). Then, using an attention parameter for each neuron and different attentional mechanisms, we simulated the response of each neuron to the different task conditions in our experiment. Finally, we assessed the population response by averaging neural responses.”

- Some of the simulation results seem to be algebraic (univariate; Fig. 7; multivariate, gain model; Fig. 8)

This is correct. We have used algebraic equations for the effect of attention on neural responses in the simulations. In fact, thinking about the two models of gain and tuning shift leads to the algebraic equations, which in turn logically leads to the observed results, if no noise is added to the data. The simulations are helpful for visualizing these logical conclusions. Also, after assigning different noise levels to each condition for each neuron, the results are not algebraic anymore which is shown in updated Figure 7 and Figure 8.

- Cross-validation does not seem to be employed - strong/weak categories seem to be assigned based on the same data used for computing DVs of interest - to minimize the potential for circularity in analyses, it would be better to define preferred categories using separate data from that used to quantify - perhaps using a cross-validation scheme? This appears to be implemented in Reddy et al. (2009), a paper implementing a similar multivariate method and cited by the authors (their ref 6).

Thank you for pointing out the missing details about how we used cross-validation. In the univariate analysis, we did use cross validation, defining preferred categories and calculating category distance on one half of the data and calculating the univariate shift on the other half of the data. Similarly, we employed cross-validation for the multivariate analysis by using one half of the data to calculate the multivariate distance between category pairs, and the other half of the data to calculate the weight shift for each category pair. We have now added this methodological information in the revised manuscript.

- Multivariate distance metric - why is correlation/cosine similarity used instead of something like Euclidean or Mahalanobis distance? Correlation/cosine similarity is scale-invariant, so changes in the magnitude of the vector would not change distance, despite this likely being an important data attribute to consider.

Since we are considering response patterns as vectors in each ROI, there is no major difference between the two measures for similarity. Using euclidean distance as a measure of distance (i.e. inverse of similarity) we observed the same relationship between weight shift and category euclidean distance. There was a positive correlation between weight shift and the euclidean category distance in all ROIs (ps < 0.01, ts > 2.9) except for V1 (p = 0.5, t = 0.66). We include this information in the revised manuscript in the Results section lines 513-515:

“We also calculated category distance based on the euclidean distance between response patterns of category pairs and observed a similarly positive correlation between the weight shift and the euclidean category distance in all ROIs (ps < 0.01, ts >2.9) except V1 (p = 0.5, t = 0.66).”

- Details about simulations implemented (and their algebraic results in some cases) make it challenging to interpret or understand these results. E.g., the noise properties of the simulated data aren't disclosed, nor are precise (or approximate) values used for simulating attentional modulations.

We clarify that the average response to each category was based on previous neurophysiology studies (Ni et al., 2012, Bao and Tsao, 2018). The attentional parameter was also chosen based on previous neurophysiology (Ni et al., 2012) and human fMRI (Doostani et al., 2023) studies of visual attention by randomly assigning a value in the range from 1 to 10. We have included the details in the Methods section in lines 357-366:

“We simulated the action of the response gain model and the tuning sharpening model using numerical simulations. We composed a neural population of 4⨯105 neurons in equal proportions body-, car-, cat- or house-selective. Each neuron also responded to object categories other than its preferred category, but to a lesser degree and with variation. We chose neural responses to each stimulus from a normal distribution with the mean of 30 spikes/s and standard deviation of 10 and each neuron was randomly assigned an attention factor in the range between 1 and 10 using a uniform distribution. These values are comparable with the values reported in neural studies of attention and object recognition in the ventral visual cortex (Ni et al. 2012, Bao and Tsao 2018). We also added poisson noise to the response of each neuron (Britten et al. 1993), assigned randomly for each condition of each neuron.”

- Eye movements do not seem to be controlled nor measured. Could it be possible that some stimulus pairs result in more discriminable patterns of eye movements? Could this be ruled out by some aspect of the results?

Subjects were instructed to direct their gaze towards the fixation point. Given the variation in the pose and orientation of the stimuli, it is unlikely that eye movements would help with the task. Eye movements have been controlled in previous experiments with individual stimulus presentation (Xu and Vaziri-Pashkam, 2019) and across attentional tasks in which colored dots were superimposed on the stimuli (Vaziri-Pashkam and Xu, 2017) and no significant difference for eye movement across categories or conditions was observed. As such, we do not think that eye movements would play a role in the results we are observing here.

- A central, and untested/verified, assumption is that the multivariate activation pattern associated with 2 overlapping stimuli (with one attended) can be modeled as a weighted combination of the activation pattern associated with the individual stimuli. There are hints in the univariate data (e.g., Fig. 4C; 4D) that this might not be justified, which somewhat calls into question the interpretability of the multivariate results.

If the reviewer is referring to the higher response in the paired compared to the isolated conditions, as explained above, we have not forced any limit on the sum of the estimated weights to equal 1 or 2. Therefore, our model is an estimation of a linear combination of the two multivariate patterns in the isolated conditions. In fact, Leila Reddy et al. (reference 6) reported that while the combination is closer to a weighted average than to a weighted sum, the sum of the weights are on average larger than 1. In Figure 4C and 4D the responses in the paired conditions are higher than either of the isolated-condition responses. This suggests that the weights for the linear combination of isolated responses in the multivariate analysis should add up to larger than one. This is what we find in our results. We have added a supplementary figure to Figure 6, depicting the sum of weights for different category pairs in all ROIs. The figure illustrates that in each ROI, the sum of weights are greater than 1 for some category pairs. It is however noteworthy that we normalized the weights in each condition by the sum of weights to calculate the weight shift in our analysis. The amount of the weight shift was therefore not affected by the absolute value of the weights.

- Throughout the manuscript, the authors consistently refer to "tuning sharpening", an idea that's almost always used to reference changes in the width of tuning curves for specific feature dimensions (e.g., motion direction; hue; orientation; spatial position). Here, the authors are assaying tuning to the category (across exemplars of the category). The link between these concepts could be strengthened to improve the clarity of the manuscript.

The reviewer brings up an excellent point. Whereas tuning curves have been extensively used for feature dimensions such as stimulus orientation or motion direction, here, we used the term to describe the variation in a neuron’s response to different object stimuli.

With a finite set of object categories, as is the case in the current study, the neural response in object space is discrete, rather than a continuous curve illustrated for features such as stimulus orientation. However, since more preferred and less preferred features (objects in this case) can still be defined, we illustrated the neural response using a hypothetical curve in object space in Figure 3 to show how it relates with other stimulus features. Therefore, here, tuning sharpening refers to the fact that the response to the more preferred object categories has been enhanced while the response to the less preferred stimulus categories is suppressed.

We clarify this point in the revised manuscript in the Discussion section lines 649-659:

“While tuning curves are commonly used for feature dimensions such as stimulus orientation or motion direction, here, we used the term to describe the variation in a neuron’s response to different object stimuli. With a finite set of object categories, as is the case in the current study, the neural response in object space is discrete, rather than a continuous curve illustrated for features such as stimulus orientation. The neuron might have tuning for a particular feature such as curvature or spikiness (Bao et al., 2020) that is present to different degrees in our object stimuli in a continuous way, but we are not measuring this directly. Nevertheless, since more preferred and less preferred features (objects in this case) can still be defined, we illustrate the neural response using a hypothetical curve in object space. As such, here, tuning sharpening refers to the fact that the response to the more preferred object categories has been enhanced while the response to the less preferred stimulus categories is suppressed.”

**Recommendations for the authors:**

**Reviewer #1 (Recommendations For The Authors):**
a. The authors should address the apparent paradox noted above (and report whether it is seen in other regions of interest as well). On what model would the response to any pair of stimuli exceed that of the response to the preferred stimulus alone? This implies some kind of Gestalt interaction whereby the combined pair generates a percept that is even more effective for the voxels in question than the "most preferred" one?

The response to a pair of stimuli can exceed the response to each of the stimuli presented in isolation if the voxel is responsive to both stimuli and as long as the voxel has not reached its saturation level. This phenomenon has been reported in many previous studies (Zoccolan et al., 2005, Reddy et al., 2009, Ni et al., 2012, Doostani et al., 2023) and can be modeled using a linear combination model which does not limit the weights of the isolated responses to equal 1 (Doostani et al., 2023). Note that the “most preferred” stimulus does not necessarily saturate the voxel response, thus the response to two stimuli could be more effective based on voxel responsiveness to the second stimulus.

As for the current study, the labels “more preferred” and “less preferred” are only relatively defined (as explained in the Methods section), meaning that the more preferred stimulus is not necessarily the most preferred stimulus for the voxels. Furthermore, the presented stimuli are semi-transparent and presented with low-contrast, which moves the responses further away from the saturation level. Based on reported evidence for multiple-stimulus responses, responses to single stimuli are in many cases sublinearly added to yield the multiple-stimulus response (Zoccolan et al., 2005, Reddy et al., 2009, Doostani et al., 2023). This means that the multiple-stimulus response is lower than the sum of the isolated responses and not lower than each of the isolated responses. Therefore, it is not paradoxical to observe higher responses in paired conditions compared to the isolated conditions. We observe similar results in other ROIs, which we provide as supplementary figures to Figure 4 in the revised manuscript.

We address this observation and similar reports in previous studies in the Results section of the revised manuscript in lines 409-413:

“Note that the response in paired conditions can be higher or lower than the response to the isolated more preferred stimulus (condition Mat), depending on the voxel preference for the two presented stimuli, as previously reported (Doostani et al., 2023). This is consistent with previous studies reporting the response to multiple stimuli to be higher than the average, but lower than the sum of the response to isolated stimuli (Reddy et al., 2009).”

b. Paradox aside, I wondered to what extent the results are in part explained by range limits. Take two categories that evoke a highly similar response (either mean over a full ROI, or in the multivariate sense). That imposes a range limit such that attentional modulation, if it works the way we think it does, could only move responses within that narrow range. In contrast, the starting point for two highly dissimilar categories leaves room in principle for more modulation.

We do not believe that the results can be explained by range limits because responses in paired conditions are not limited by the isolated responses, as can be observed in Figure 4. However, to rule out the possibility of the similarity between responses in isolated conditions affecting the range within which responses in paired conditions can change, we turned to the multivariate analysis. We used the weight shift measure as the change in the weight of each stimulus with the change in the attentional target. In this method, no matter how close the two isolated vectors are, the response to the pair could still have a whole range of different weights of the isolated responses. We have plotted an example illustration of two-dimensional vectors for better clarification. Here, the vectors Vxat and Vyat denote the responses to the isolated x and y stimuli, respectively, and the vector Pxaty denotes the response to the paired condition in which stimulus x is attended. The weights a1 and a2 are illustrated in the figure, which are equal to regression coefficients if we solve the equation Pxaty = [a1 a2] [x y]’. While the weight values depend on the amplitude of and the angle between the three vectors, they are not limited by a lower angle between Vxat and Vyat.

We have updated Figure 2 in the manuscript to avoid the confusion. We have also added a figure including the sum of weights for different category pairs in different regions, showing that the sum of weights are not dependent on the similarity between the two stimuli. The conclusions based on the weight shift are therefore not confounded by the similarity between the two stimuli.

c. Finally, related to the previous point, while including V1 is a good control, I wonder if it is getting a "fair" test here, because the range of responses to the four categories in this region, in terms of (dis)similarity, seems compressed relative to the other categories.

We believe that V1 is getting a fair test because the single-subject range of category distance in V1 is similar to LO, as can be observed Author response image 1_:_

**Author response image 1. sa2fig1:** Range of category distance in each ROI averaged across participants.

The reason that V1 is showing a more compressed distance range on the average plot is that the category distance in V1 is not consistent among participants. Although the average plots are shown in Figure 5 and Figure 6, we tested statistical significance in each ROI based on single-subject correlation coefficients.

Please also note that a more compressed range of dissimilarity does not necessarily lead to a less strong effect of category distance on the effect of attention. For instance, while LO shows a more compressed dissimilarity range for the presented categories compared to the other object selective regions, it shows the highest correlation between weight shift and category distance. Furthermore, as illustrated in Figure 5, no significant correlation is observed between univariate shift and category distance in V1, even though the range of the univariate distance in V1 is similar to LO and pFs, where we observed a significant correlation between category distance and univariate shift.

d. In general, the manuscript does a very good job explaining the methods of the study in a way that would allow replication. In some places, the authors could be clearer about the reasoning behind those methodological choices. For example: - How was the sample size determined?

Estimating conservatively based on the smallest amount of attentional modulation we observed in a previous study (Doostani et al., 2023), we chose a medium effect size (0.3). For a power of 0.8, the minimum number of participants should be 16. We have added the explanation to the Methods section in lines 78-81:

“We estimated the number of participants conservatively based on the smallest amount of attentional modulation observed in our previous study (Doostani et al., 2023). For a medium effect size of 0.3 and a power of 0.8, we needed a minimum number of 16 participants.”

- Why did the authors choose those four categories? What was the evidence that would suggest these would span the range of similarities needed here?

We chose these four categories based on a previous behavioral study reporting the average reaction time of participants when detecting a target from one category among distractors from another category (Xu and Vaziri-Pashkam, 2019). Ideally the experiment should include as many object categories as possible. However, since we were limited by the duration of the experiment, the number of conditions had to be controlled, leading to a maximum of 4 object categories. We chose two animate and two inanimate object categories to include categories that are more similar and more different based on previous behavioral results (Xu and Vaziri-Pashkam, 2019). We included body and house categories because they are both among the categories to which highly responsive regions exist in the cortex. We chose the two remaining categories based on their similarity to body and house stimuli. In this way, for each category there was another category that elicited similar cortical responses, and two categories that elicited different responses. While we acknowledge that the chosen categories do not fully span the range of similarities, they provide an observable variety of similarities in different ROIs which we find acceptable for the purposes of our study.

We include this information in the Methods section of the revised manuscript in lines 89-94:

“We included body and house categories because there are regions in the brain that are highly responsive and unresponsive to each of these categories, which provided us with a range of responsiveness in the visual cortex. We chose the two remaining categories based on previous behavioral results to include categories that provided us with a range of similarities (Xu and Vaziri-Pashkam, 2019). Thus, for each category there was a range of responsiveness in the brain and a range of similarity with the other categories.”

- Why did the authors present the stimuli at the same location? This procedure has been adopted in previous studies, but of course, it does also move the stimulus situation away from the real-world examples of cluttered scenes that motivate the Introduction.

We presented the stimuli at the same location because we aimed to study the mechanism of object-based attention and this experimental design helped us isolate it from spatial attention. We do not think that our design moves the stimulus situation away from real-world examples in such a way that our results are not generalizable. We include real-world instances, as well as a discussion on this point, in the Discussion section of the revised manuscript, in lines 611-620:

“Although examples of superimposed cluttered stimuli are not very common in everyday life, they still do occur in certain situations, for example reading text on the cellphone screen in the presence of reflection and glare on the screen or looking at the street through a patterned window. Such instances recruit object-based attention which was the aim of this study, whereas in more common cases in which attended and unattended objects occupy different locations in space, both space-based and object-based attention may work together to resolve the competition between different stimuli. Here we chose to move away from usual everyday scenarios to study the effect of object-based attention in isolation. Future studies can reveal the effect of target-distractor similarity, i.e. proximity in space, on space-based attention and how the effects caused by object-based and space-based attention interact.”

- While I'm not concerned about this (all relevant comparisons were within-participants) was there an initial attempt to compare data quality from the two different scanners?

We compared the SNR values of the two groups of participants and observed no significant difference between these values (ps > 0.34, ts < 0.97). We have added this information to the Methods section.

Regarding the observed effect, we performed a t-test between the results of the participants from the two scanners. For the univariate results, the observed correlation between univariate attentional modulation and category distance was not significantly different for participants of the two scanners in any ROIs (ps > 0.07 , ts < 1.9). For the multivariate results, the observed correlation between the weight shift and multivariate category distance was not significantly different in any ROIs (ps > 0.48 , ts < 0.71) except for V1 (p-value = 0.015 , t-value = 2.75).

We include a sentence about the comparison of the SNR values in the preprocessing section in the revised manuscript.

e. There are a couple of analysis steps that could be applied to the existing data that might strengthen the findings. For one, the authors have adopted a liberal criterion of p < 0.001 uncorrected to include voxels within each ROI. Why, and to what extent is the general pattern of findings robust over more selective thresholds? Also, there are additional regions that are selective for bodies (fusiform body area) and scenes (occipital place area and retrosplenial cortex). Including these areas might provide more diversity of selectivity patterns (e.g. different responses to non-preferred categories) that would provide further tests of the hypothesis.

We selected this threshold to allow for selection of a reasonable number of voxels in each hemisphere across all participants. To check whether the effect is robust over more selective thresholds, we exemplarily redefined the left EBA region using p < 0.0001 and p < 0.00001 and observed that the weight shift effect remained equivalent. We have made a note of this analysis in the Results section. As for the additional regions suggested by the reviewer, we chose not to include them because they could not be consistently defined in both hemispheres of all participants. Please note that the current ROIs also show different responses to non-preferred categories (e.g. in LO and pFs). We include this information in the Methods section in lines 206-207:

“We selected this threshold to allow for selection of a reasonable number of voxels in each hemisphere across all participants.”

And in the Results section in lines 509-512:

“We performed the analysis including only voxels that had a significantly positive GLM coefficient across the runs and observed the same results. Moreover, to check whether the effect is robust over more selective thresholds for ROI definition, we redefined the left EBA region with p < 0.0001 and p < 0.00001 criteria. We observed a similar weight shift effect for both criteria.”

f. One point the authors might address is the potential effect of blocking the paired conditions. If I understood right, the irrelevant item in each paired display was from the same category throughout a block. To what extent might this knowledge shape the way participants attend to the task-relevant item (e.g. by highlighting to them certain spatial frequencies or contours that might be useful in making that particular pairwise distinction)? In other words, are there theoretical reasons to expect different effects if the irrelevant category is not predictable?

We believe that the participants’ knowledge about the distractor does not significantly affect our results because our results are in agreement with previous behavioral data (Cohen et al., 2014, Xu and Vaziri-Pashkam, 2019), in which the distractor could not be predicted. These reports suggest there is a theoretical reason to expect similar effects if the participants could not predict the distractor. To directly test this, one would need to perform an fMRI experiment using an event-related design, an interesting venue for future research.

We have made a note of this point in the Discussion section of the revised manuscript in lines 621-626:

“Please note that we used a blocked design in which the target and distractor categories could be predicted across each block. While it is possible that the current design has led to an enhancement of the observed effect, previous behavioral data (Cohen et al., 2014, Xu and Vaziri-Pashkam, 2019) have reported the same effect in experiments in which the distractor was not predictable. To study the effect of predictability on fMRI responses, however, an event-related design is more appropriate, an interesting venue for future fMRI studies.”

g. The authors could provide behavioural data as a function of the specific category pairs. There is a clear prediction here about which pairs should be more or less difficult.

We provide the behavioral data as a supplementary figure to Figure 1 in the revised manuscript. We however do not see differences in behavior for the different category paris. This is so because our fMRI task was designed in a way to make sure the participants could properly attend to the target for all conditions. The task was rather easy across all conditions and due to the ceiling effect, there was no significant difference between behavioral performance for different category pairs. However, the effect of category pair on behavior has been previously tested and reported in a visual search paradigm with the same categories (Xu and Vaziri-Pashkam, 2019), which was in fact the basis for our choice of categories in this study (as explained in response to point “d” above).

h. Figure 4 shows data for EBA in detail; it would be helpful to have a similar presentation of the data for the other ROIs as well.

We provide data for all ROIs as figure supplements 1-4 to Figure 4 in the revised manuscript.

i. For the pFs and LOC ROIs, it would be helpful to have an indication of what proportion of voxels was most/least responsive to each of the four categories. Was this a relatively even balance, or generally favouring one of the categories?

In LO, the proportion of voxels most responsive to each of the four categories was relatively even for Body (31%) and House (32%) stimuli, which was higher than the proportion of Car- and Cat-preferring voxels (18% and 19%, respectively). In pFs, 40% of the voxels were house-selective, while the proportion was relatively even for voxels most responsive to bodies, cars, and houses with 21%, 17%, and 22% of the voxels, respectively. We include the percentage of voxels most responsive to each of the four categories in each ROI as Appendix 1-table 1.

j. Were the stimuli in the localisers the same as in the main experiment?

No, we used different sets of stimuli for the localizers and the main experiment. We have added the information in line 146 of the Methods section.

**Reviewer #2 (Recommendations For The Authors):**
(1) Why are specific ROIs chosen? Perhaps some discussion motivating these choices, and addressing the possible overlap between these and retinotopic regions (based on other studies, or atlases - Wang et al, 2015) would be useful.

Considering that we used object categories, we decided to look at general object-selective regions (LO, pFS) as well as regions that are highly selective for specific categories (EBA, PPA). We also looked at the primary visual cortex as a control region. We have added this clarification in the Methods section lines 128-133:

“Considering that we used object categories, we investigated five different regions of interest (ROIs): the object-selective areas lateral occipital cortex (LO) and posterior fusiform (pFs) as general object-selective regions, the body-selective extrastriate body area (EBA) and the scene-selective parahippocampal place area (PPA) as regions that are highly selective for specific categories, and the primary visual cortex (V1) as a control region. We chose these regions because they could all be consistently defined in both hemispheres of all participants and included a large number of voxels.”

(2) The authors should consider including data on the relative prevalence of voxels preferring each category for each ROI (and/or the mean activation level across voxels for each category for each ROI). If some ROIs have very few voxels preferring some categories, there's a chance the observed results are a bit noisy when sorting based on those categories (e.g., if a ROI has essentially no response to a given pair of categories, then there's not likely to be much attentional modulation detectable, because the ROI isn't driven by those categories to begin with).

We thank the reviewer for the insightful comment.

We include the percentage of voxels most responsive to each of the four categories in each ROI in the Appendix (Appendix 1-table 1, please see the answer to point “i” of the first reviewer).

We also provide a table of average activity across voxels for each category in all ROIs as Appendix 1-table 2.

As shown in the table, voxels show positive activity for all categories in all ROIs except for PPA, where voxels show no response to body and cat stimuli. This might explain why we observed a marginally significant correlation between weight shift and category distance in PPA only. As the reviewer mentions, since this region does not respond to body and cat stimuli, we do not observe a significant change in response due to the shift in attention for some pairs. We include the table in the Appendix and add the explanation to the Results section of the revised manuscript in lines 506-508:

_“_Less significant results in PPA might arise from the fact that PPA shows no response to body and cat stimuli and little response to car stimuli (Appendix 1-table 2). Therefore, it is not possible to observe the effect of attention for all category pairs.”

a. Related - would it make sense to screen voxels for inclusion in analysis based on above-basely activation for one or both of the categories? [could, for example, imagine you're accidentally measuring from the motor cortex - you'd be able to perform this analysis, but it would be largely nonsensical because there's no established response to the stimuli in either isolated or combined states].

We performed all the analyses including only voxels that had a significantly positive GLM coefficient across the runs and the results remained the same. We have added the explanation in the Results section in line 509-510.

(3) Behavioral performance is compared against chance level, but it doesn't seem that 50% is chance for the detection task. The authors write on page 4 that the 1-back repetition occurred between 2-3 times per block, so it doesn't seem to be the case that each stimulus had a 50% chance of being a repetition of the previous one.

We apologize for the mistake in our report. We have reported the detection rate for the target-present trials (2-3 per block), not the behavioral performance across all trials. We have modified the sentence in the Results section.

(4) Authors mention that the stimuli are identical for 2-stimulus trials where each category is attended (for a given pair) - but the cue is different, and the cue appears as a centrally-fixated word for 1 s. Is this incorporated into the GLM? I can't imagine this would have much impact, but the strict statement that the goals of the participant are the only thing differentiating trials with otherwise-identical stimuli isn't quite true.

The word cue was not incorporated as a separate predictor into the GLM. As the reviewer notes, the signals related to the cue and stimuli are mixed. But given that the cues are brief and in the form of words rather than images, they are unlikely to have an effect on the response in the regions of interest.

To be more accurate, we have included the clarification in the Methods section in lines 181-182:

“We did not enter the cue to the GLM as a predictor. The obtained voxel-wise coefficients for each condition are thus related to the cue and the stimuli presented in that condition.”

And in the Results section in lines 425-428 :

“It is important to note that since the cue was not separately modeled in the GLM, the signals related to the cue and the stimuli were mixed. However, given that the cues were brief and presented in the form of words, they are unlikely to have an effect on the responses observed in the higher-level ROIs.”

(5) Eq 5: I expected there to be some comparison of a and b directly as ratios (e.g., a_1 > b_1, as shown in Fig. 2). The equations used here should be walked through more carefully - it's very hard to understand what this analysis is actually accomplishing. I'm not sure I follow the explanation of relative weights given by the authors, nor how that maps onto the delta_W quantity in Equation 5.

We provide a direct comparison of a and b, as well as a more thorough clarification of the analysis, in the Methods section in lines 274-276:

“We first projected the paired vector on the plane defined by the isolated vectors (Figure 2A) and then determined the weight of each isolated vector in the projected vector (Figure 2B).”

And in lines 286-297:

“A higher a1 compared to a2 indicates that the paired response pattern is more similar to Vxat compared to Vyat, and vice versa. For instance, if we calculate the weights of the Body and Car stimuli in the paired response related to the simultaneous presentation of both stimuli, we can write in the LO region: VBodyatCar = 0.81 VBody + 0.31 VCar, VBodyCarat = 0.43 VBody + 0.68 VCar. Note that these weights are averaged across participants. As can be observed, in the presence of both body and car stimuli, the weight of each stimulus is higher when attended compared to the case when it is unattended. In other words, when attention shifts from body to car stimuli, the weight of the isolated body response (VBody) decreases in the paired response. We can therefore observe that the response in the paired condition is more similar to the isolated body response pattern when body stimuli are attended and more similar to the isolated car response pattern when car stimuli are attended.”

And lines 303-306:

“As shown here, even when body stimuli are attended, the effect of the unattended car stimuli is still present in the response, shown in the weight of the isolated car response (0.31). However, this weight increases when attention shifts towards car stimuli (0.68 in the attended case).”

We also provide more detailed clarification for the 𝛥w and the relative weights in lines 309-324:

“To examine whether this increase in the weight of the attended stimulus was constant or depended on the similarity of the two stimuli in cortical representation, we defined the weight shift as the multivariate effect of attention:

𝛥w = a1/(a1+a2) – b1/(b1+b2) (5)

Here, a1, a2, b1,and b2 are the weights of the isolated responses, estimated using Equation 4. We calculate the weight of the isolated x response once when attention is directed towards x (a1), and a second time when attention is directed towards y (b1). In each case, we calculate the relative weight of the isolated x in the paired response by dividing the weight of the isolated x by the sum of weights of x and y (a1+a2 when attention is directed towards x, and b1+b2 when attention is directed towards y). We then define the weight shift, Δw, as the change in the relative weight of the isolated x response in the paired response when attention shifts from x to y. A higher Δw for a category pair indicates that attention is more efficient in removing the effect of the unattended stimulus in the pair. We used relative weights as a normalized measure to compensate for the difference in the sum of weights for different category pairs. Thus, using the normalized measure, we calculated the share of each stimulus in the paired response. For instance, considering the Body-Car pair, the share of the body stimulus in the paired response was equal to 0.72 and 0.38, when body stimuli were attended and unattended, respectively. We then calculated the change in the share of each stimulus caused by the shift in attention using a simple subtraction (Equation 5: Δw=0.34 for the above example of the Body-Car pair in LO) and used this measure to compare between different pairs.”

We hope that this clarification makes it easier to understand the multivariate analysis and the weight shift calculation in Equation 5.

We additionally provide the values of the weights (a1, b1, a2, and b2) for each category pair averaged across participants as Appendix 1 -table 4.

(6) For multivariate analyses (Fig. 6A-E), x axis is normalized (pattern distance based on Pearson correlation), while the delta_W does not seem to be similarly normalized.

We calculated ΔW by dividing the weights in each condition by the sum of weights in that condition. Thus, we use relative weights which are always in the range of 0 to 1, and ΔW is thus always in the range of -1 to 1. This means that both axes are normalized. Note that even if one axis were not normalized, the relationship between the independent and the dependent variables would remain the same despite the change in the range of the axis.

(7) Simulating additional scenarios like attention to both categories just increasing the mean response would be helpful - is this how one would capture results like those shown in some panels of Fig. 4?

We did not have a condition in which participants were asked to attend to both categories. Therefore it was not useful for our simulations to include such a scenario. Please also note that the goal of our simulations is not to capture the exact amount of attentional modulation, but to investigate the effect of target-distractor similarity on the change in attentional modulation (univariate shift and weight shift).

As for the results in some panels of Figure 4, we have explained the reason underlying higher responses in paired conditions compared to isolated conditions in response to the “weaknesses” section of the second reviewer. We hope that these points satisfy the reviewer’s concern regarding the results in Figure 4 and our simulations.

(8) Lines 271-276 - the "latter" and "former" are backwards here I think.

We believe that the sentence was correct, but confusing.. We have rephrased the sentence to avoid the confusion in lines 371-376 of the revised manuscript:

“We modeled two neural populations: a general object-selective population in which each voxel shows preference to a particular category and voxels with different preferences are mixed in with each other (similar to LO and pFS), and a category-selective population in which all voxels have a similar preference for a particular category (similar to EBA and PPA).”

(9) Line 314 - "body-car" pair is mentioned twice in describing the non-significant result in PPA ROI.

Thank you for catching the typo. We have changed the second Body-Car to Body-Cat.

(10) Fig. 5 and Fig. 6 - I was expecting to see a plot that demonstrated variability across subjects rather than across category pairs. Would it be possible to show the distribution of each pair's datapoints across subjects, perhaps by coloring all (e.g.) body-car datapoints one color, all body-cat datapoints another, etc? This would also help readers better understand how category preferences (which differ across ROIs) impact the results.

We demonstrated variability across category pairs rather than subjects because we aimed to investigate how the variation in the similarity between categories (i.e. category distance) affected the univariate and multivariate effects of attention. The variability across subjects is reflected in the error bars in the bar plots of Figure 5 and Figure 6.

Here we show the distribution of each category pair’s data points across subjects by using a different color for each pair:

**Author response image 2. sa2fig2:** Univariate shift versus category distance including single-subject data points in all ROIs.

**Author response image 3. sa2fig3:** Weight shift versus category distance including single-subject data points in all ROIs.

As can be observed in the figures, category preference has little impact on the results. Rather, the similarity in the preference (in the univariate case) or the response pattern (in the multivariate case) to the two presented categories is what impacts the amount of the univariate shift and the weight shift, respectively. For instance, in EBA we observe a low amount of attentional shift both for the Body-Cat pair, with two stimuli for which the ROI is highly selective, and the Car-House pair, including stimuli to which the region shows little response. A similar pattern is observed in the object-selective regions LO and pFs which show high responses to all stimulus categories.

We believe that the figures including the data points related to all subjects are not strongly informative. However, we agree that using different colors for each category pair helps the readers better understand that category preference has little impact on the results in different ROIs. We therefore present the colored version of Figure 5 and Figure 6 in the revised manuscript, with a different color for each category pair.

(11) Fig. 5 and Fig. 6 use R^2 as a dependent variable across participants to conclude a positive relationship. While the positive relationship is clear in the scatterplots, which depict averages across participants for each category pair, it could still be the case that there are a substantial number of participants with negative (but predictive, thus high positive R^2) slopes. For completeness and transparency, the authors should illustrate the average slope or regression coefficient for each of these analyses.

We concluded the positive relationship and calculated the significance in Figure 5 and Figure 6 using the correlation r rather than r.^2 This is why the result was not significantly positive in V1. We acknowledge that the use of r-squared in the bar plot leads to confusion. We have therefore changed the bar plots to show the correlation coefficient instead of the r-squared. Furthermore, we have added a table of the correlation coefficient for all participants in all ROIs for the univariate and weight shift analyses supplemental to Figure 5 and Figure 6, respectively.

(12) No statement about data or analysis code availability is provided

Thanks for pointing this out. The fMRI data is available on OSF. We have added a statement about it in the Data Availability section of the revised manuscript in line 669.